# ARC-ENCODER: LEARNING COMPRESSED TEXT REPRESENTATIONS FOR LARGE LANGUAGE MODELS

## ABSTRACT

Recent techniques such as retrieval-augmented generation or chain-of-thought reasoning have led to longer contexts and increased inference costs. Context compression techniques can reduce these costs, but the most effective approaches require fine-tuning the target model or even modifying its architecture. This can degrade its general abilities when not used for this specific purpose. Here we explore an alternative approach: an encoder that compresses the context into continuous representations which replace token embeddings in decoder LLMs. First, we perform a systematic study of training strategies and architecture choices for the encoder. Our findings led to the design of an Adaptable text Representations Compressor, named ARC-Encoder, which outputs $x$-times fewer continuous representations (typically $x \in \{4, 8\}$) than text tokens. We evaluate ARC-Encoder across a variety of LLM usage scenarios, ranging from in-context learning to context window extension, on both instruct and base decoders. Results show that ARC-Encoder achieves state-of-the-art performance on several benchmarks while improving computational efficiency at inference. Finally, we demonstrate that our models can be adapted to multiple decoders simultaneously, allowing a single encoder to generalize across different decoder LLMs. This makes ARC-Encoder a flexible and efficient solution for portable encoders that can support multiple LLMs, requiring only small model-specific projectors for adaptation.

## 1 INTRODUCTION

As their use expands, LLMs are required to process increasingly long contexts to incorporate detailed user prompts or external knowledge retrieved from large corpora as in retrieval-augmented generation (RAG) systems (Lewis et al., 2021). However, this implies a *large computational cost* at inference due to the quadratic complexity of Transformer attention mechanisms (Tay et al., 2022). Furthermore, using longer contexts can dilute interesting information leading to poor downstream results or can even reach the *context window limit* of the LLM, which damages the model capacities.

To address this issue, context compression is a promising solution. It reduces input length while preserving the semantics necessary for accurate generation (Liu et al., 2025). Techniques fall into two main categories: *hard compression*, which prunes, summarizes or deletes tokens (Jiang et al., 2023b), offering interpretability and model-agnosticism but limited compression; and *soft compression*, which encodes context into dense vectors, e.g. memory tokens and gist tokens (Mu et al., 2024; Ge et al., 2024). While achieving higher compression, soft methods often require training specialized encoders and even adapting the decoder itself (Louis et al., 2025a; Tang et al., 2025).

Our work aims at leveraging LLMs' compression ability in cases where documents should be processed on the fly. Most importantly, we constrain ourselves not to modify the decoder so that the method remains plug-and-play while maintaining a degree of flexibility between compression and accuracy on downstream tasks. In this paper, we introduce an Adaptable text Representations Compressor ("ARC-Encoder") that produces *pooled tokens*, optimized to be directly consumable by a decoder in the same way as standard input tokens (injected after the embedding matrix). By preserving the few-shot abilities of base models (Brown et al., 2020), we ensure compatibility with the standard question-answering (QA) evaluation set-up, such as 5-shot evaluation using exact match (EM). We achieve nearly the same accuracy as a decoder operating on full text, even at a $4\times$ pooling factor while using an encoder adaptable to several decoders by means of a small MLP which has

less than $1\%$ of the encoder parameters. To the best of our knowledge, our approach achieves state-of-the-art results among models that do not fine-tune the decoder. We further adapt it to context extension applications and show that it also works in a zero-shot setting using instruct models as decoders.

Here is a summary of our main contributions:

- We introduce ARC-Encoder, a method to compute compressed text representations that replace the raw text input in LLMs. Our approach reduces the input sequence length, *without requiring any modification to the decoder model*. ARC-Encoder preserves strong performance across various benchmarks and scenarios, including in-context learning (ICL).
- We show that a single encoder can be trained to work with multiple decoders, requiring less than $1\%$ of specific parameters per LLM. ARC-Encoder can be further adapted to new decoders with minimal adjustment.
- ARC-Encoder can be trained to extend a decoder context size, by compressing the chunks of a large document in parallel, showing competitive results on long-context benchmarks.
- Finally, we show that both pretraining and fine-tuning are key to the success of our approach. We also find that the compressed representations of Wikipedia require memory on the same order as that needed to store the raw text, allowing to precompute representations.

## 2 RELATED WORK

**Encoder-Decoder architectures.** Text auto-encoders have long been studied to improve transformers on specific downstream tasks. They process the input into dense embeddings to reduce processing cost while preserving or improving model accuracy. For instance, Atlas (Izacard et al., 2022) retrieves and encodes multiple relevant passages before decoding with a focus on knowledge intensive tasks. RAVEN (Huang et al., 2024) uses a similar retrieval-augmented encoder-decoder structure and improves in-context learning abilities while using less compute. More recently, Zhang et al. (2025) propose an asymmetric architecture, closer to ours, where a smaller encoder aims at improving the decoder generation through cross-attention while using less compute.

**Context Compression.** Context compression reduces the number of tokens processed by a model to improve the efficiency of inference. There are two main approaches. *Hard compression* methods, such as LLMLingua (Jiang et al., 2023b), operate directly in the text space by removing tokens from prompts. It aims at reducing their length while preserving the performance of the model. In contrast, we perform *soft compression* which involves learning continuous compressed representations. This line of work began with gist tokens (Mu et al., 2024), which summarize task instructions into a few tokens by modifying the attention matrix to force generated tokens to only attend to gist tokens. Similarly, in summary vectors or memory tokens (Chevalier et al., 2023; Ge et al., 2024), learnable vectors which come from an encoder are prepended to the input sequence. These vectors serve as condensed representations of the full sequence when passed through the decoder. These methods typically rely on a pretraining phase to align the encoder's output with the decoder's hidden states, followed by fine-tuning of both encoder and decoder to fully leverage the compressed representations (Louis et al., 2025a;b). Recently, more similarly to our pooling method, Tang et al. (2025) explore using merged tokens to replace memory ones. They perform several training stages on the encoder, but also on the decoder in contrast to our work. Fine-tuning the decoder often degrades performance on standard tokens compared to our method which enables to use compressed tokens as well as standard ones. Other approaches explore the use of pre-computed text embeddings as memory tokens, reaching higher pooling factors (up to $\times 150$) with xRAG (Cheng et al., 2024) but performing poorly on certain benchmarks and lacking compression flexibility. All these context compression methods rely on the intrinsic compression capacity of LLMs. Indeed, Kuratov et al. (2025) has proven that an LLM decoder can be used to directly compress a text passage of roughly 1568 tokens into just one 4096-dimensional vector. More recently, Eyuboglu et al. (2025) leveraged this property to produce sets of compressed KV-caches for frequently used long documents. Recently, alternative methods use vision encoders (Xing et al., 2025) to produce compressed textual representations from text rendered as image.

**Long Context.** Recent work on long-context language modeling combines fine-tuning with extended positional encoding strategies. Together AI (2023) fine-tunes Llama2 7B to handle 32k-token

inputs using position interpolation (Chen et al., 2023). Zhang et al. (2024) extends context length by inserting activation "anchors" into the hidden states of the model, requiring modification and fine-tuning of the target LLM itself to compress. In contrast, Yen et al. (2024) introduce a lightweight encoder that processes long inputs in parallel and passes compressed representations to a decoder via learned cross-attention, allowing efficient long-sequence handling. Similar works such as Han et al. (2024) rely on additional chunking strategy in addition of the encoder to also compress information. Our work follows the idea of Yen et al. (2024) while producing fewer tokens and keeping the decoder untouched.

## 3 METHOD

### 3.1 ARCHITECTURE

The overall architecture that we consider comprises a text *encoder* and an *MLP projector*, together forming a trainable ARC-Encoder, followed by the frozen target *decoder*. The encoder is based on an LLM transformer, from which we remove the output head and the causal mask. We add a pooling mechanism that reduces the number of elements in the sequence from $n$ to $\frac{n}{x}$ with $x$ the pooling factor (PF). The MLP projector is a 2-layer MLP without activation, mapping the encoder output to the embedding dimension of the decoder through a dimensional bottleneck. The decoder remains unchanged, as opposed to the encoder and MLP that are trained. The compressed continuous representations from the ARC-Encoder are used instead of the token embeddings in the decoder.

### 3.2 POOLING METHOD

Most context compression works use learned or memory tokens as compressed representations. This makes it harder to compress well sequences of various sizes, as the number of output representations is fixed. Instead, we pool hidden state vectors directly, leading to a fixed pooling factor, independent of the input sequence length (Suganthan et al., 2025).

We performed an empirical study on how and where to pool tokens to obtain compressed continuous representations. As illustrated in Fig. 1, pooling is performed in the self-attention module. We average consecutive queries to reach the targeted pooling factor, while keys and values remain unchanged. For a PF of 2 for example (denoting the encoder as 'ARC$_2$-Encoder'), we group the tokens of the sequence two-by-two. We merge their queries in the last self-attention module, by averaging their continuous hidden states. Then, these pooled queries attend the non-compressed keys and values, mimicking a standard self-attention, but with two-times fewer queries, resulting in a pooling factor of two. We explored inserting the pooling mechanism earlier in the encoder, but this lead to poorer performance. This follows the intuition that the information should be as processed as possible before pooling (Tang et al., 2025).

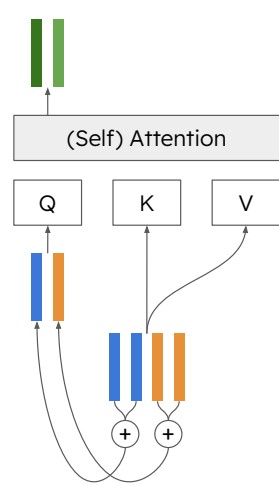

Figure 1: **Pooling in ARC$_2$-Encoder**. In the last SA module, queries are merged by pairs of successive tokens.

### 3.3 TRAINING

**Base pretraining.** Memory tokens (Ge et al., 2024) show the importance of the reconstruction task when training the model to create easily decompressible continuous representations. However, pure reconstruction is easier than compressing contexts for downstream tasks. In fact, with proper training, our auto-encoder architecture achieves near-perfect reconstruction at a × pooling factor on relatively short sequences (up to around 128 tokens). Unfortunately, these compressed representations cannot be well exploited by the decoder on downstream tasks, as the model tends to regurgitate the entire context, instead of extracting pertinent information from it. Thus, we consider a second pretraining task, continuation, which is better aligned with inference-time behavior. It consists in replacing subsequences of natural text by their compressed representations, and to teacher-force the continuation immediately following compressed segments. We alternate between these two pretrain-

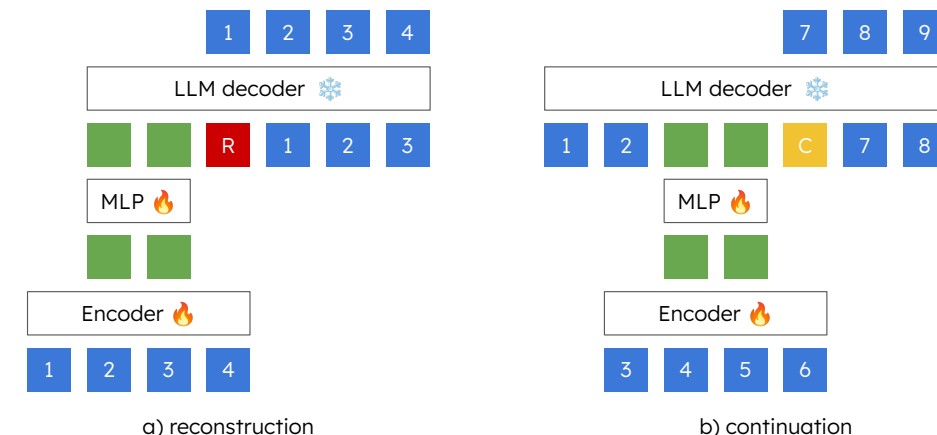

Figure 2: **ARC-Encoder pretraining tasks**. The encoder, special tokens and the MLP are trained through two alternating tasks: a) Reconstruction: compressed tokens are given to the decoder which is teacher-forced to replicate the full text tokens; b) Continuation: a subpart of the tokens in the sequence are compressed and the decoder is teacher-forced to continue starting from the partially compressed sequence. Illustration for ARC$_2$-Encoder.

ing tasks as summarized in Fig. 2, and use the standard cross-entropy loss. Additionally, we append special learned tokens <Cont> and <Rec> after each compressed sequence. The two special tokens are crucial to specify the task during pretraining and enable good downstream results.

**Fine-Tuning.** After pretraining, our ARC-Encoder can be flexibly adapted to a range of specific tasks. In particular, it can be used to condition a frozen decoder on task-specific inputs, allowing task specialization without modifying the decoder itself. To preserve the decoder's few-shot capabilities, we optionally include a small number of in-context examples in the continuation objective– interleaving compressed documents, full-text queries and answers–following a structured prompt template described in Appendix C. To compute the loss, we mask all tokens except the ones that continue the last compressed sequence, corresponding to the final answer in a few-shot setting. This design encourages generalization and preserves ICL possibilities at inference time with every document being compressed. Furthermore, we demonstrate that zero-shot abilities of instruct decoders can also be preserved; it simply requires to use a different pretraining and fine-tuning template. Once pretrained, ARC-Encoder can be fine-tuned to master in-context learning, long context understanding, or other downstream tasks. The pooling factor between the two training stages can be changed. It leads to improved models when pretraining with a higher factor than for fine-tuning, see Appendix A.2. We remind that only the encoder is fine-tuned, and that the decoder is unchanged.

**Multi-Decoder Training.** To improve the generality of our method, we design a compressor capable of generating token representations that can be used by multiple decoders without any modification. It is a nontrivial challenge due to inherent discrepancies between the hidden-state spaces of different decoder architectures. To overcome this, we employ a shared encoder but we specialize a projector layer (the MLP in Fig. 2) and the special tokens. This set of learned parameters accounts for less than $1\%$ of the encoder's weights, enabling decoder-specific adaptation with minimal overhead. We find that training with an alternating objective provides the most stable and effective results. At each training step we sample a decoder uniformly and update only its associated projector as well as the shared encoder. This strategy ensures balanced exposure across decoders while maintaining the generalization capacity of the shared encoder.

## 4 EXPERIMENTS

In this section, we first describe the main settings used for our experiments. Second, we evaluate our method on various downstream tasks that use contexts, such as question answering with retrieved documents or reading comprehension. The contexts for these tasks tend to be short, ranging from 30 to 1,500 tokens. Thus, we also evaluate ARC-Encoders for long-context understanding applications.

Finally, we report the results of our ablation study and an analysis of the memory used to encode Wikipedia with continuous representations from ARC-Encoder.

## 4.1 EXPERIMENTAL SETTING

**Models.** We perform experiments using three different decoders: Llama3.1 8B (Grattafiori et al., 2024), Mistral 7B (Jiang et al., 2023a), both base models, as well as Llama2 7B Chat (Touvron et al., 2023). We design our ARC-Encoder with Llama3.2 3B as the backbone: we remove its last two layers and no causal mask is applied. The MLP consists of two layers: the first projects the 3072-dimensional vectors from the encoder to 2048 dimensions, and the second projects these vectors to 4096-dimensional hidden states, matching the decoder dimension. During inference, we append the special <Cont> token after every sequence of compressed tokens.

**Training & Datasets.** By default, we decide to train all layers of the encoder, including the embedding matrix, using AdamW optimizer (Loshchilov & Hutter, 2019). For pretraining, we use data from Common Crawl that has been filtered and processed using dactory[1], keeping samples with a quality score higher than $0.2$. ARC-Encoder is pretrained on approximately 2.6B tokens. For fine-tuning, we use two different mixes of synthetic and supervised datasets: one for standard context compression benchmarks using base models as decoders and another for long-context benchmarks using an instruct decoder. See the following sections and Appendix C.3 for more details.

## 4.2 CONTEXT COMPRESSION

**Benchmarks.** We evaluate our method on question answering, translation, and summarization tasks. For question answering, we use HotpotQA (Yang et al., 2018, HQA) with the *distractor* setting, Natural Questions (Kwiatkowski et al., 2019, NQ), TriviaQA (Joshi et al., 2017, TQA) and SQuAD (Rajpurkar et al., 2016). Top-5 passages are retrieved using NV-Embed v2 (Lee et al., 2025) from Atlas (Izacard et al., 2022) Wikipedia chunks, simulating a RAG setup. We report exact match (EM) as the main metric, where $EM = 1$ if the normalized predicted and reference answers exactly match. For translation, we evaluate on FLORES (Goyal et al., 2021) averaging BLEU scores across four directions: English to Danish, French, German, and Spanish. Summarization performance is evaluated on CNN-DailyMail using ROUGE-L, which aligns well with human judgments for summarization abilities according to Zhang et al. (2023). All models are evaluated via 5-shot with compressed contexts. The reported pooling factor reflects the per-context ratio of original tokens (using the decoder tokenizer) to compressed tokens. See Appendix D.2 for full evaluation details.

**Setting & Baselines.** Unlike prior works in the context compression literature that often report zero-shot accuracy with instruct decoders—potentially inflating performance when models simply replicate the context—our use of EM aims at better capturing real-world LLM usage by rewarding answers following the few-shot format patterns. We believe it better measures the encoder abilities to produce useful representations, from which the decoder can extract information. Thus, we focus on base model decoders for context compression evaluations. We first set two baselines that reflect the decoder intrinsic abilities. In the first one, denoted *closed-book*, the decoder relies only on its parametric knowledge and in the second one, called *open-book*, the decoder has access to uncompressed documents in its context.

For meaningful comparisons, we select a diverse set of strong baselines that capture different approaches to context compression. These include: i) LLMLingua2 (Pan et al., 2024), which performs hard compression; ii) ICAE (Ge et al., 2024), a soft compression approach using memory tokens; iii) xRAG (Cheng et al., 2024), which relies on pre-computed embeddings for retrieval-augmented generation; iv) PISCO (Louis et al., 2025a), an approach close to ICAE which fine-tunes both the encoder and decoder and states that pretraining is not necessary. We re-implemented the last three baselines using our decoder, fine-tuning dataset, and interleaved fine-tuning task format to ensure a consistent few-shot evaluation setup while preserving each method's core design. This allows direct comparisons with a common evaluation protocol; See Appendix D.1 for implementation details.

We fine-tune models on a mix of synthetic translation data and supervised datasets (QA, summarization, paraphrasing, and reading comprehension), explicitly excluding the training sets of our

---

[1] https://github.com/kyutai-labs/dactory

Table 1: **Main comparison of ARC-Encoder and other models**. 'PF' (pooling factor): the token reduction factor (e.g., $4\times$) for fixed-ratio methods or the number of compressed tokens used, e.g. $\sim 16$, when this number is fixed as this yields benchmark-dependent ratios; 'Param.': number of parameters of the encoder; 'Avg. length': mean number of tokens per context document. The superscript on ARC-Encoder indicates if the model is specifically trained for one decoder ($^M$ for Mistral or $^L$ for Llama) or both simultaneously ($^\otimes$). $^\dagger$ marks modified re-implementations, see details in Appendix D.1. Best context compression results are in **bold**, second best are underlined.

| | Methods | PF | Param. | NQ | TQA | HQA | SQuAD | FLORES | CNN | Avg. |
|---|---|---|---|---|---|---|---|---|---|---|
| | Avg. length | | | 155 | 152 | 1479 | 185 | 30 | 956 | |
| *Mistral 7B decoder* | *closed-book* | $\infty$ | | 29.1 | 62.4 | 22.8 | 17.1 | | | |
| | *open-book* | $1\times$ | | 39.9 | 70.5 | 48.3 | 77.7 | 31.3 | 27.2 | 49.2 |
| | ICAE-like$^\dagger$ | $\sim 32$ | 7.2B | 36.5 | 66.7 | 24.3 | 58.8 | 28.3 | 15.8 | 38.4 |
| | xRAG-like$^\dagger$ | $\sim 1$ | 7.1B | 30.7 | 65.2 | 21.5 | 23.9 | 0.9 | 14.6 | 26.1 |
| | LLMLingua2 | $1.9\times$ | 0.6B | 38.8 | 69.0 | 43.7 | 59.2 | 12.6 | 24.9 | 41.4 |
| | PISCO-like$^\dagger$ | $\sim 32$ | 7.2B | 34.7 | 68.5 | 24.9 | 38.2 | 33.6 | 19.2 | 36.5 |
| | | $4\times$ | – | 36.6 | 69.2 | 29.4 | 48.1 | **34.5** | 19.3 | 39.5 |
| | ARC$_4$-Encoder$^\otimes$ | $4\times$ | 3.0B | 38.2 | **70.4** | 40.8 | 69.2 | 29.5 | **25.6** | 45.6 |
| | ARC$_4$-Encoder$^M$ | $4\times$ | – | **39.0** | 68.9 | **45.1** | **71.1** | 31.0 | 23.8 | **46.5** |
| | ARC$_8$-Encoder$^M$ | $8\times$ | – | 38.4 | 67.9 | 40.8 | 62.0 | 28.3 | 22.9 | 43.4 |
| | Avg. length | | | 135 | 133 | 1285 | 164 | 27 | 855 | |
| *Llama3.1 8B decoder* | *closed-book* | $\infty$ | | 25.4 | 60.6 | 21.6 | 15.3 | | | |
| | *open-book* | $1\times$ | | 38.6 | 67.1 | 47.1 | 72.2 | 32.8 | 26.5 | 47.4 |
| | ICAE-like$^\dagger$ | $\sim 32$ | 7.6B | 38.4 | 67.3 | 20.5 | 61.6 | 31.3 | 17.3 | 39.4 |
| | xRAG-like$^\dagger$ | $\sim 1$ | 7.1B | 28.0 | 62.1 | 21.7 | 22.3 | 3.4 | 12.7 | 25.0 |
| | LLMLingua2 | $2.0\times$ | 0.6B | 36.1 | 66.3 | 45.2 | 58.8 | 13.6 | 23.8 | 40.6 |
| | PISCO-like$^\dagger$ | $\sim 32$ | 7.6B | 35.1 | 69.4 | 30.6 | 40.5 | 35.2 | 19.7 | 38.4 |
| | | $4\times$ | – | 37.9 | 70.5 | 37.0 | 57.2 | **36.5** | 20.7 | 43.3 |
| | ARC$_4$-Encoder$^\otimes$ | $4\times$ | 3.0B | 39.6 | **70.8** | 43.6 | 71.8 | 32.8 | **26.1** | 47.5 |
| | ARC$_4$-Encoder$^L$ | $4\times$ | – | **39.7** | 70.1 | **46.9** | **74.0** | 33.7 | 23.7 | **48.0** |
| | ARC$_8$-Encoder$^L$ | $8\times$ | – | 38.9 | 69.0 | 42.8 | 66.0 | 30.6 | 22.8 | 45.0 |

evaluation benchmarks. This setup better highlights our method's generalization ability (using the train sets of benchmarks strongly improves results, sometimes outperforming the *open-book* results as shown in Tab. 6 of the Appendix). Since each data sample is drawn from one of the sub-datasets, fine-tuning involves stochasticity; unless otherwise specified, we fix the random seed to 0. For our final ARC-Encoder we follow the best pretraining / fine-tuning pooling factor pairs as shown in Fig. 6. This leads to ARC$_4$-Encoder using the same pretrained encoder as ARC$_8$-Encoder, only with its fine-tuning performed at a pooling factor of 4.

**Results.** In Tab. 1, we report our main results on context compression using the best checkpoints of ARC-Encoder. First, we observe that our models outperform the *closed-book* baseline, showing that frozen decoders can extract useful information for downstream tasks from the compressed tokens. We also note that ARC-Encoder tends to outperform the baselines more on tasks where the decoder cannot rely on its own parametric memory, such as reading comprehension (SQuAD) or summarization (CNN). Furthermore, we observe that both specialized and shared ARC-Encoder perform better when paired with the Llama3.1 8B decoder, likely because it belongs to the same model family as our encoder backbone based on Llama3.2 3B. Finally, ARC$_4$-Encoder nearly matches the *open-book* baseline without altering the decoder and while achieving $\times 1.8$ gains of prefilling FLOPs, as profiled in Appendix B. Please note that for certain baselines, our results differ from the ones reported in previous work. This is due to the different setting that we consider in this paper, namely 1) using exact match as the metric, 2) excluding the training sets of the benchmarks from the fine-tuning data and 3) using base models instead of instruct ones.

**Encoder adaptation to multi-decoder.** Through multiple experiments, we found that a single encoder can be trained to be used by multiple decoders. More specifically, during both pretraining

and fine-tuning, we use a joint learning setup where at each training step, we sample which decoder to use among the two targets. To improve performance, we introduce a separate MLP projector for each decoder. This allows lightweight specialization of compressed representations, adding only 15M parameters per decoder. We report results in Tab. 1, showing that in average, the common encoder, ARC-Encoder$^\otimes$, looses less than $1.0$ point compared to its specialized counterparts. More encoder-decoder pairings are tested in Appendix A.3.

**Adaptation to new decoders.** Once our ARC-Encoder$^\otimes$ has been trained to work with two decoders, we can adapt it to new decoders with minimal adjustments. Indeed, we solely learn a new MLP projector and special tokens to feed a third decoder, OLMo-7B (Groeneveld et al., 2024), while keeping the encoder frozen. Fine-tuning leads to better results than the *closed-book* baseline by only training 15M parameters. However, the score gap with the *open-book* setting remains relatively larger than when training with the alternating decoder objective as shown in Tab. 1 with Llama and Mistral models. Interestingly, on benchmarks where the decoder was limited by its context window, such as HotpotQA, using ARC$_4$-Encoder outperforms the *open-book* setup.

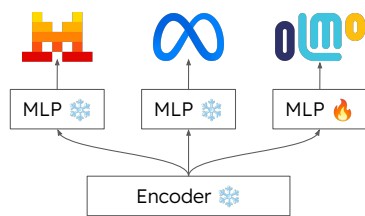

Figure 3: Extending an ARC-Encoder to a new decoder.

Table 2: **Adaptation to a new decoder**. Due to OLMo 7B 2048-token context window, we truncate documents for *open-book* baseline to $400$ tokens. Per decoder specific parameters are reported.

| | Methods | PF | Param. | **NQ** | **TQA** | **HQA** | **SQuAD** | **FLORES** | **CNN** | **Avg.** |
|---|---|---|---|---|---|---|---|---|---|---|
| *OLMo-7B* | *closed-book* | $\infty$ | | 19.5 | 48.3 | 17.8 | 11.2 | | | |
| | *open-book* | $1\times$ | | 35.3 | 64.8 | 22.9 | 67.9 | 22.2 | 24.2 | 39.6 |
| | ARC$_4$-Encoder$^M$ | $4\times$ | 15M | 31.5 | 62.5 | 26.5 | 46.4 | 17.2 | 19.1 | 33.9 |
| | ARC$_4$-Encoder$^\otimes$ | $4\times$ | – | 33.1 | 63.1 | 25.0 | 44.6 | 17.1 | 18.9 | 33.6 |

### 4.3 LONG CONTEXT

In this section, we adapt our fine-tuning method to handle longer contexts, testing our architecture on long-context understanding tasks. We pretrain and fine-tune an ARC$_8$-Encoder paired with an instruct decoder Llama2 7B Chat (Touvron et al., 2023), and refer the reader to Appendix C for technical details. During fine-tuning, we encode fixed-size chunks of long documents in parallel and feed them to the decoder by concatenating the compressed tokens of each chunk. For fine-tuning, we synthesize a QA and summarization dataset based on concatenated Wikipedia chunks, PG-19 books (Rae et al., 2019) and ArXiv papers from RedPajama (Weber et al., 2024) using Gemma3-27B (Team et al., 2025). Then, we divide each context in up to 32 chunks of $1024$ tokens. Similarly at inference, contexts are truncated to 32k tokens and then ARC$_8$-Encoder processes chunks in parallel. In this setting, we remove the special tokens, as instruction prompts play a similar role.

**Benchmarks & Models.** For long-context understanding, we report F1 score on NarrativeQA and QASPER and report Rouge-L on GovReport and QM-Sum validation sets from the ZeroSCROLLS benchmark (Shaham et al., 2023), a suite of zero-shot long-context understanding tasks. Specifically, we adopt the task formats and instructions as used in Yen et al. (2024)[2]. On these benchmarks, we compare our model to Llama2 7B Chat, which is constrained by a context window of $4096$ tokens, as well as to open-source models specifically designed to extend its limited context window. These include Llama2-32k Instruct (Together AI, 2023), which relies on positional interpolation combined with fine-tuning, and CEPED (Yen et al., 2024), which employs a lightweight encoder to process chunks of input in parallel, feeding their representations into a decoder through learned cross-attention layers.

**Results.** In Tab. 3, we show that feeding compressed tokens from ARC$_8$-Encoder to Llama2 Chat substantially improves long-context QA performance. It allows the model to process up to $8\times$ more input than its original context window. This also shows that the decoder can interpret the

---

[2]https://github.com/princeton-nlp/CEPE

Table 3: **Long-context evaluation on long-context benchmarks**. The token count includes all tokens fed to the decoder. CEPED uses 2k decoder tokens plus encoder-side tokens.

| Models | Max. Tokens | NQA | Qspr | GvRp | QM-Sum |
|---|---|---|---|---|---|
| Llama2 Chat | 4k | 16.1 | 17.2 | 15.7 | 19.8 |
| + CEPED | 2k + 30k | 20.5 | 19.7 | 12.7 | 19.7 |
| Llama2-32k Instruct | 32k | 14.2 | 16.4 | 17.8 | 17.6 |
| $ARC_8$-Encoder + Llama2 Chat | 4k (32k//8) | 27.5 | 28.3 | 14.1 | 19.1 |

compressed tokens without any parameter modification. Thus, a small model's context window can be extended simply by training an external compressing encoder. On some tasks, our approach even outperforms methods that expand Llama2's context window through new learned internal modules or full-model fine-tuning. This advantage may stem partly from our synthesized fine-tuning dataset, which matches the answer-length distributions of evaluation benchmarks, as other models do not fine-tune specifically on these tasks. In Appendix A.7, we show that ARC-Encoder achieves strong and more consistent fine-grained retrieval performance across varying context lengths than other tested models. Crucially, ours is the only approach that leaves the decoder unchanged, ensuring identical behavior across all other tasks while improving the decoder on long-context understanding.

## 4.4 COMPRESSION ABLATIONS

In this section, we discuss key design choices of our method. We compare context compression results using the same evaluation setting as in Tab. 1, mostly showing the average score on all these benchmarks. Unless stated otherwise, models are pretrained and then fine-tuned using a pooling factor of 8 with the Mistral 7B decoder. To reduce the computation costs of these ablations we pretrain on approximately 2B tokens only, roughly 75% of the tokens used for models in Tab. 1. Additional details are provided in Appendix C.1.

**Training objective.** In contrast to Louis et al. (2025a), we show that pretraining is essential for our approach. While fine-tuning is also crucial, it cannot provide competitive results on downstream tasks on its own. Long pretraining is essential for aligning ARC-Encoder outputs with the decoder's hidden state space. For example, after 20k pretraining steps, we observe an improvement of approximately +16 points on the average score, while after 80k steps, the improvement reaches +19 points, compared to directly fine-tuning without pretraining. Without fine-tuning the decoder fails to use the compressed context leading to large performance drop in translations, reading comprehension and summarization.

Table 4: Impact of pretraining reconstruction ratio.

| % **Rec.** | **Avg.** |
|---|---|
| 0% | 39.8 |
| 20% | **41.6** |
| 50% | 41.5 |
| 100% | 37.5 |

Beyond pretraining length, the choice of pretraining tasks also proves critical: Tab. 4 shows that omitting reconstruction or using too little continuation leads to substantial performance drops. In practice, early training updates mainly reduce the reconstruction loss, suggesting that the encoder and MLP first learn to produce correctly aligned compressed representations with the embedding space of the decoder, after which the continuation objective encourages representations that the LLM can use effectively for downstream generation. We also tried adding context distillation as in Cheng et al. (2024) during fine-tuning but it did not improve results while adding a large computational overhead.

**Encoder Architecture.** This ablation explores design choices for reducing encoder size or boosting performance. Thanks to strong results in multi-decoder and long-context applications, we keep the default setting architecture in Tab. 1 for consistency reasons, though other designs may excel in specific use cases. Tab. 5 shows the trade-off between encoder parameters and performance when truncating layers of the LLM backbone. Alternative pooling schedules also look promising: pooling every two tokens in the last layers performs better than our default at the same pooling factor, suggesting pooling strategies could be adapted to the target pooling factor. In addition, removing causality, which is effective only when training all layers improves encoder capacity. Adding special learned tokens is particularly useful to help encoder generalize to new decoders.

Table 5: **Ablations on encoder design**. *Default setting* corresponds to $ARC_8$-$Encoder^M$ with only 60k pretraining steps. All results are averaged over 3 fine-tunings with different seeds.

| | Param. | NQ | TQA | HQA | SQuAD | FLORES | CNN | Avg. |
|---|---|---|---|---|---|---|---|---|
| *Default setting* | 3.0B | 36.9 | 67.2 | 39.9 | 58.3 | 27.4 | 20.1 | 41.7 |
| *How to truncate the encoder?* | | | | | | | | |
| Truncate 0 layer | 3.2B | 38.8 | 67.7 | 39.4 | 61.6 | 26.9 | 20.3 | 42.4 |
| Truncate 4 layers | 2.8B | 37.6 | 67.5 | 39.1 | 60.1 | 25.0 | 20.4 | 41.6 |
| Truncate 21 layers | 1.1B | 37.0 | 67.3 | 32.5 | 52.1 | 23.5 | 19.2 | 38.6 |
| *How to pool?* | | | | | | | | |
| by 2 every last layers | 3.0B | 38.2 | 68.4 | 40.0 | 61.1 | 26.7 | 20.1 | 42.4 |
| by 2 every two layers | – | 38.3 | 68.1 | 38.9 | 61.1 | 26.4 | 20.0 | 42.1 |
| *How to modify the encoder?* | | | | | | | | |
| w LoRA (rank= 128) | – | 38.8 | 67.0 | 37.0 | 57.7 | 27.0 | 19.1 | 41.1 |
| w causality | – | 38.7 | 67.9 | 37.4 | 57.3 | 27.0 | 19.4 | 41.3 |

**Pooling.** The pooling operation should merge information from continuous representations while still producing vectors interpretable by the decoder. We experiment with memory tokens, the standard approach in the field. This method performs poorly as sequence length increases (see Tab. 8 in appendix), since the effective compression becomes higher. We also test clustering queries with k-means and averaging those within the same cluster, but this merging of potentially distant tokens proved harmful especially in translation tasks. It assumes in addition a non-causal training of the encoder, failing completely without it. The most effective poolings instead use contiguous tokens, either by averaging them as in Fig. 1, or by selecting the last token of each segment. Since averaging proved more robust in the multi-decoder setting, we choose it as our default method.

## 4.5 MEMORY ANALYSIS

When compressing contexts on the fly, $ARC_4$-Encoder already leads to a $1.8\times$ speed-up compared to using the natural text. In the case where contexts are potentially used multiple times, such as in RAG systems, even greater speed-ups could be achieved by *pre-computing* the compressed representations and storing them. This option is only viable if the size of compressed representations is roughly the same as the original text. Here, we explore the following tradeoffs to reduce the size of compressed representations: 1) changing the pooling factor, 2) reducing the dimension of the MLP bottleneck and 3) quantizing the representations using product quantization (Jégou et al., 2011, PQ), by increasing the dimension of the sub-quantizers while keeping the number of centroids fixed. We report results in Fig. 4, showing that by combining these different methods (the bottleneck dimension varies within curves of the same color, and the marker shape indicates the pooling factor), encoding English Wikipedia with ARC-Encoder requires 80 GiB with minimal impact on performance, or 20 GiB while still improving the closed book baseline significantly. For comparison, the raw text of English Wikipedia requires approximately 24 GiB, thus making ARC-Encoder suitable for pre-computing compressed representations.

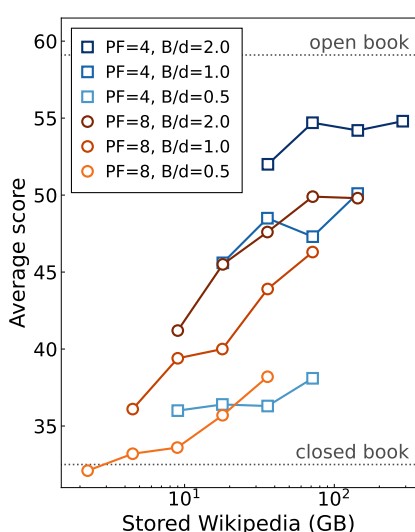

Figure 4: Compression results with varying MLP dimensional bottlenecks and number of bits per dimension (B/d).

---

[2] https://github.com/facebookresearch/faiss/wiki

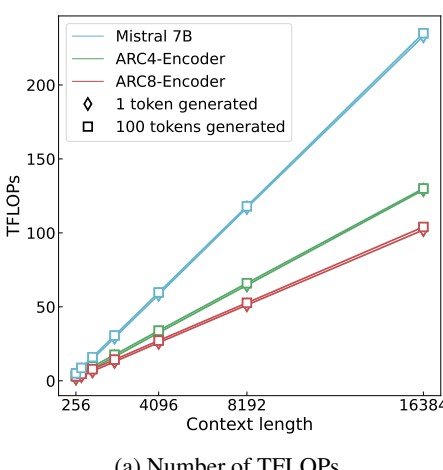 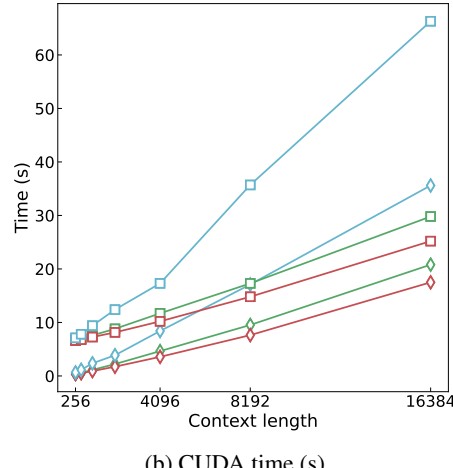

(a) Number of TFLOPs          (b) CUDA time (s)

Figure 5: **Measured computational costs**. (a) Number of TFLOPs and (b) CUDA time in seconds for the continuation of a book from PG19 for various prompt lengths and numbers of tokens to generate on one NVIDIA H100.

### 4.6 PROFILING

By compressing context we aim at speeding up inference as well as reducing the computations. In practice, we evaluate these benefits using Torch Profiler[3] measuring CUDA execution time (s) and tera FLOPs (TFLOPS) for the compression, prefilling and decoding stages with various prompt context lengths and various numbers of tokens to generate. All experiments use Mistral 7B decoder and are run in float32 on one NVIDIA H100 GPU with a batch size of 1. We force the decoder to continue its context prompt, compressed or not, and generate $n \in \{1, 100\}$ tokens. As shown in Fig. 5, generation is less costly in terms of compute when using compressed tokens by ARC-Encoder. The compute cost of compression is amortized during the prefilling phase since the decoder has fewer tokens to process.

## 5 CONCLUSION

We introduce ARC-Encoder, a novel method to compute compressed text representations that can replace the raw text input in large language models. By reducing the context length, our method leads to faster prefilling and decoding stages, while leaving the target LLM unchanged. We show that a single encoder can be trained to work with multiple decoders, or even extended to new decoders with minimal adaptation. This opens the way towards *universal compressed representations*. We show that pretraining and fine-tuning are both critical for the success of our approach. In terms of architecture, pooling in the attention mechanism leads to strong results, while allowing a constant pooling factor for different sequence sizes, as opposed to memory tokens. Finally, using an MLP between the encoder and decoder allows our approach to compress representations further and to learn a single encoder for multiple decoders.

### REPRODUCIBILITY STATEMENT

We took particular care to provide all technical details in both the main text and the appendix to reproduce our experiments. We plan to release a code-base for training and evaluating ARC-Encoder, along with pretrained ARC-Encoder checkpoints and the fine-tuning dataset.

We acknowledge the use of LLM-based tools (e.g., those integrated in Overleaf) to help reformulate and polish the writing of this paper.

---

[3]`https://docs.pytorch.org/tutorials/recipes/recipes/profiler_recipe.html`

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

# A FURTHER EXPERIMENTS

## A.1 CONTEXT COMPRESSION WITH BENCHMARK TRAIN SETS

To demonstrate the generalization ability of ARC-Encoder, we deliberately avoid fine-tuning on the training sets of evaluation benchmarks. However, we believe that this could be an interesting use case for users who wish to specialize ARC-Encoder in a given domain. In Tab. 6, we report results from models fine-tuned on our dataset augmented with the HotpotQA and SQuAD training sets. These results show that, even without altering decoders, ARC-Encoders can specialize effectively on specific benchmarks. They outperform the *open-book* baseline while maintaining efficiency gains and without harming performance on other tasks.

Table 6: **Performance when adding HotpotQA and SQuAD train sets to the fine-tuning dataset.** 'PF': the token reduction factor (e.g., $4\times$) for fixed-ratio methods or the number of compressed tokens used, e.g. $\sim 16$, when this number is fixed; 'Param.': number of parameters of the encoder; The superscript on ARC-Encoder indicates if the model is specifically trained for one decoder ($^M$ for Mistral or $^L$ for Llama) or both simultaneously ($^\otimes$). Best context compression results are in **bold**, second best are underlined.

| | Method | PF | Param. | NQ | TQA | HQA | SQuAD | FLORES | CNN | Avg. |
|---|---|---|---|---|---|---|---|---|---|---|
| **Mistral 7B** | *closed-book* | $\infty$ | | 29.1 | 62.4 | 22.8 | 17.1 | | | |
| | *open-book* | $1\times$ | | 39.9 | 70.5 | 48.3 | 77.7 | 31.3 | 27.2 | 49.2 |
| | ARC$_4$-Encoder$^\otimes$ | $4\times$ | 3.0B | 38.3 | **68.9** | 60.5 | 77.0 | 29.9 | **26.0** | 50.1 |
| | ARC$_4$-Encoder$^M$ | $4\times$ | – | 38.4 | 67.9 | **60.7** | **81.1** | 30.9 | 22.7 | **50.3** |
| | ARC$_8$-Encoder$^M$ | $8\times$ | – | **39.0** | 67.0 | 57.5 | 74.8 | 28.0 | 20.8 | 47.9 |
| **Llama3.1 8B** | *closed-book* | $\infty$ | | 25.4 | 60.6 | 21.6 | 15.3 | | | |
| | *open-book* | $1\times$ | | 38.6 | 67.1 | 47.1 | 72.2 | 32.8 | 26.5 | 47.4 |
| | ARC$_4$-Encoder$^\otimes$ | $4\times$ | 3.0B | 38.5 | **69.7** | 61.9 | 78.6 | **33.2** | **26.0** | 51.3 |
| | ARC$_4$-Encoder$^L$ | $4\times$ | – | **40.7** | 68.5 | **62.1** | **82.3** | **33.2** | 22.2 | **51.5** |
| | ARC$_8$-Encoder$^L$ | $8\times$ | – | 38.6 | 67.6 | 59.1 | 76.0 | 30.1 | 21.7 | 48.9 |

## A.2 POOLING FACTOR GENERALIZATION

Fig. 6 demonstrates that ARC-Encoder pretrained at a certain pooling factor can still be fine-tuned at another, sometimes even improving results on downstream tasks. This transfer works best as we use a smaller pooling factor than the one previously pretrained on. Notably, pretraining at $8\times$ seems to be particularly effective since we can then outperform any other pair on pooling factors of $4\times$ and $8\times$. In contrast, models do not generalize well when fine-tuned to higher pooling factors. This capability greatly benefits the method since it enables to reach better results at various pooling factors while pretraining fewer models. We train our best ARC$_4$-Encoder and ARC$_8$-Encoder from models pretrained using a pooling factor of $8$. When pooling too much performance degrades sharply as with a pooling factor of $32$ where the model has an averaged score of 33.1.

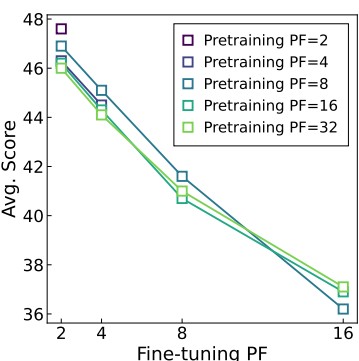

Figure 6: Score for various pairs of pooling factor between pretraining and fine-tuning.

## A.3 ENCODER-DECODER PAIRS

Following the same recipe as described in Section 3, we can design ARC-Encoder / decoder pairs based on other backbone models. Tab. 7 shows that Llama3.1 8B can also serve as an encoder backbone, displaying higher affinity with Llama decoders, similar to Llama3.2 3B. Adding the MLP projector is crucial to adapt to different decoders in the multi-decoder case as well as in the specific decoder one. Our experiments further reveal that the hidden state spaces of Llama3.1 8B and Mistral

7B are not fully disentangled: a single encoder can feed both decoders with the same compressed representations while outperforming the closed-book baseline. We attribute this compatibility to their similar architectures and training pipelines.

Table 7: **Different backbones for an ARC$_4$-Encoder paired with different decoders**. The score is the average of the metrics in the Tab. 1. For underlined modules, the weights are the same for the two decoders; they are decoder-specific otherwise.

| Decoder | Encoder | | | | |
|---|---|---|---|---|---|
| | L8B | L3B + MLP | L8B | L8B + MLP | L3B + MLP |
| *Llama3.1 8B* | 46.0 | 47.2 | 32.4 | 46.2 | 45.5 |
| *Mistral 7B* | 44.8 | 45.1 | 32.8 | 41.8 | 43.2 |

## A.4 POOLING METHODS

Table 8: **Different pooling methods**. Scores are averaged over 3 fine-tunings with different seeds.

| | Pooling method | PF | NQ | TQA | HQA | SQuAD | FLORES | CNN | Avg. |
|---|---|---|---|---|---|---|---|---|---|
| *Mistral 7B* | Average queries | − | 36.9 | 67.2 | 39.9 | 58.3 | 27.4 | 20.1 | 41.7 |
| | Last queries | − | 37.6 | 68.1 | 39.7 | 59.8 | 27.3 | 19.7 | 42.1 |
| | Kmeans merged queries | − | 37.1 | 65.8 | 38.4 | 52.3 | 18.6 | 19.4 | 38.6 |
| | Memory tokens | ∼ 32 | 36.8 | 66.8 | 28.9 | 49.8 | 31.0 | 17.0 | 38.4 |

## A.5 FURTHER BASELINES EVALUATIONS

Table 9: **Further baselines evaluations.** Same setting and specifics as in Tab. 1

| | Methods | PF | Param. | NQ | TQA | HQA | SQuAD | FLORES | CNN | Avg. |
|---|---|---|---|---|---|---|---|---|---|---|
| *Mistral 7B decoder* | *closed-book* | ∞ | | 29.1 | 62.4 | 22.8 | 17.1 | | | |
| | *open-book* | 1× | | 39.9 | 70.5 | 48.3 | 77.7 | 31.3 | 27.2 | 49.2 |
| | ICAE-like[†] | ∼ 32 | 7.2B | 36.5 | 66.7 | 24.3 | 58.8 | 28.3 | 15.8 | 38.4 |
| | xRAG-like[†] | ∼ 1 | 7.1B | 30.7 | 65.2 | 21.5 | 23.9 | 0.9 | 14.6 | 26.1 |
| | | ∼ 8 | − | 28.7 | 62.4 | 21.8 | 24.2 | 9.0 | 10.7 | 26.1 |
| | LLMLingua2 | 1.9× | 0.6B | 38.8 | 69.0 | 43.7 | 59.2 | 12.6 | 24.9 | 41.4 |
| | | 3.6× | − | 35.2 | 66.6 | 36.0 | 42.0 | 4.3 | 22.1 | 34.4 |
| | PISCO-like[†] | ∼ 32 | 7.2B | 34.7 | 68.5 | 24.9 | 38.2 | 33.6 | 19.2 | 36.5 |
| | | 4× | − | 36.6 | 69.2 | 29.4 | 48.1 | **34.5** | 19.3 | 39.5 |
| | ARC$_4$-Encoder$^{\otimes}$ | 4× | 3.0B | 38.2 | **70.4** | 40.8 | 69.2 | 29.5 | **25.6** | 45.6 |
| | ARC$_4$-Encoder$^{M}$ | 4× | − | **39.0** | 68.9 | **45.1** | **71.1** | 31.0 | 23.8 | **46.5** |
| | ARC$_8$-Encoder$^{M}$ | 8× | − | 38.4 | 67.9 | 40.8 | 62.0 | 28.3 | 22.9 | 43.4 |
| *Llama3.1 8B decoder* | *closed-book* | ∞ | | 25.4 | 60.6 | 21.6 | 15.3 | | | |
| | *open-book* | 1× | | 38.6 | 67.1 | 47.1 | 72.2 | 32.8 | 26.5 | 47.4 |
| | ICAE-like[†] | ∼ 32 | 7.6B | 38.4 | 67.3 | 20.5 | 61.6 | 31.3 | 17.3 | 39.4 |
| | xRAG-like[†] | ∼ 1 | 7.1B | 28.0 | 62.1 | 21.7 | 22.3 | 3.4 | 12.7 | 25.0 |
| | | ∼ 8 | − | 26.6 | 61.1 | 21.9 | 22.3 | 6.4 | 12.6 | 25.2 |
| | LLMLingua2 | 2.0× | 0.6B | 36.1 | 66.3 | 45.2 | 58.8 | 13.6 | 23.8 | 40.6 |
| | | 3.9× | − | 34.2 | 66.1 | 37.2 | 41.9 | 3.2 | 21.3 | 34.0 |
| | PISCO-like[†] | ∼ 32 | 7.6B | 35.1 | 69.4 | 30.6 | 40.5 | 35.2 | 19.7 | 38.4 |
| | | 4× | − | 37.9 | 70.5 | 37.0 | 57.2 | **36.5** | 20.7 | 43.3 |
| | ARC$_4$-Encoder$^{\otimes}$ | 4× | 3.0B | 39.6 | **70.8** | 43.6 | 71.8 | 32.8 | **26.1** | 47.5 |
| | ARC$_4$-Encoder$^{L}$ | 4× | − | **39.7** | 70.1 | **46.9** | **74.0** | 33.7 | 23.7 | **48.0** |
| | ARC$_8$-Encoder$^{L}$ | 8× | − | 38.9 | 69.0 | 42.8 | 66.0 | 30.6 | 22.8 | 45.0 |

### A.5.1 ICAE FURTHER COMPARISONS

ICAE-style context compression relies on learned tokens (memory tokens) appended at the end of the context to compress. It produces for each encoded sequence a fixed number of compressed representations corresponding to the output of these memory tokens. In contrast, our method requires no additional learned tokens: the encoder directly pools context tokens, yielding a variable number of compressed representations that scales with input length. It enables consistent results across sequence lengths. Additionally, ICAE encoders mirror the architecture of their target decoder, while ARC-Encoders are trained starting from any decoder-only LLM (typically Llama 3.2 3B) with removed layers and bidirectional attention, making them noticeably smaller and more versatile. ICAE-like model in Tab. 1 represents an ICAE context compression architecture under the same setting as our models training and evaluation. However, direct comparison is imperfect because ICAE's pooling factor is not fixed: e.g. 32 memory tokens always produce 32 outputs, whereas ARC4-Encoder produces outputs 4x shorter than the input. To clarify this we have also trained and evaluated ICAE architectures on fixed-size chunks as described in Section 3.3.3 of the ICAE paper (Ge et al., 2024), so that they operate with a fixed pooling factor. We show downstream results of these models in Tab. 10 as well as the ICAE-like architecture with varying pooling factors across benchmarks. The downstream results show that our method outperforms ICAE-like models at matched pooling factors while requiring less than twice the encoding compute. For some benchmarks we still outperform ICAE-like models using ARC-Encoders with higher pooling factors.

Table 10: **Further baselines evaluations.** Same setting and specifics as in Tab. 1 with Mistral 7B decoder. Average pooling factors for each benchmark are indicated as subscripts of the reported scores. These variations occur when the number of memory tokens is fixed, and ICAE-like models are not trained or evaluated on fixed-size chunks, resulting in a pooling factor that depends on the context length.

| Methods | PF | NQ | TQA | HQA | SQuAD | FLORES | CNN | Avg. |
|---|---|---|---|---|---|---|---|---|
| *closed-book* | $\infty$ | 29.1 | 62.4 | 22.8 | 17.1 | | | 38.4 |
| *open-book* | $1\times$ | 39.9 | 70.5 | 48.3 | 77.7 | 31.3 | 27.2 | 49.2 |
| ICAE-like[†] | $\sim 32$ | $36.5_{(5\times)}$ | $66.7_{(5\times)}$ | $24.3_{(46\times)}$ | $58.8_{(6\times)}$ | $28.3_{(1\times)}$ | $15.8_{(32\times)}$ | 38.4 |
| | $4\times$ | 36.4 | 66.7 | 23.8 | 60.5. | 28.7 | 18.6 | 39.1 |
| | $\sim 16$ | $35.7_{(10\times)}$ | $66.7_{(9\times)}$ | $26.0_{(92\times)}$ | $51.0_{(12\times)}$ | $26.9_{(2\times)}$ | $14.3_{(64\times)}$ | 20.6 |
| | $8\times$ | 34.8 | 66.6 | 8.9 | 53.0 | 26.7 | 17.5 | 34.6 |
| | $\sim 8$ | $34.9_{(19\times)}$ | $65.3_{(19\times)}$ | $25.3_{(185\times)}$ | $28.9_{(23\times)}$ | $18.5_{(4\times)}$ | $17.7_{(128\times)}$ | 18.1 |
| | $16\times$ | 33.7 | 65.6 | 0.1 | 31.1 | 19.0 | 14.7 | 27.4 |
| ARC$_2$-Encoder[M] | $2\times$ | **41.7** | **69.3** | **48.3** | **76.7** | 30.4 | 18.9 | **47.6** |
| ARC$_4$-Encoder[M] | $4\times$ | 39.0 | 68.9 | 45.1 | 71.1 | **31.0** | **23.8** | 46.5 |
| ARC$_8$-Encoder[M] | $8\times$ | 38.4 | 67.9 | 40.8 | 62.0 | 28.3 | 22.9 | 43.4 |
| ARC$_{16}$-Encoder[M] | $16\times$ | 35.4 | 67.1 | 31.8 | 45.1 | 22.7 | 20.3 | 37.1 |
| ARC$_{32}$-Encoder[M] | $32\times$ | 34.6 | 65.8 | 28.8 | 34.8 | 17.0 | 17.8 | 33.1 |

### A.5.2 GENERALIZATION ON OUT-OF-DOMAIN PROFESSIONAL DATASETS

To assess the generalization ability of our fine-tuned models on strictly out-of-domain benchmarks, we evaluate the above models on BioASQ (Krithara et al., 2023) and PubMedQA (Jin et al., 2019). These two QA datasets focus on biological and biomedical question-answering. Both domains do not appear the fine-tuning datasets enabling to test out-of-domain generalization. It further highlights the strengths of the following models in specialized professional domains. Since all compressor models we compare against, except for LLMLingua2, were fine-tuned on the same dataset, they are expected to struggle with the out-of-domain nature of the provided contexts. In shown in Tab. 11, our ARC-Encoder achieves the best performance even with a pooling factor of $8$, reaching results close to the upper-bound *open-book* setting. These findings demonstrate the strong real-world applicability and robustness of our approach.

Table 11: **Out-of-domain evaluations.** Same setting and specifics as in Tab. 1. Both are evaluated using the F1 metric.

| | Methods | PF | Param. | **PubMedQA** | **BioASQ** |
|---|---|---|---|---|---|
| *Mistral 7B decoder* | *closed-book* | $\infty$ | | 58.7 | 63.6 |
| | *open-book* | $1\times$ | | 84.4 | 77.5 |
| | ICAE-like$^{\dagger}$ | $\sim 32$ | 7.2B | 63.6 | 57.7 |
| | LLMLingua2 | $1.9\times$ | 0.6B | 74.4 | 73.4 |
| | | $3.6\times$ | – | 66.4 | 69.7 |
| | PISCO-like$^{\dagger}$ | $\sim 32$ | 7.2B | 62.4 | 60.5 |
| | | $4\times$ | – | 62.0 | 66.0 |
| | ARC$_4$-Encoder$^{\otimes}$ | $4\times$ | 3.0B | 73.9 | 72.0 |
| | ARC$_4$-Encoder$^{M}$ | $4\times$ | – | 76.6 | **75.6** |
| | ARC$_8$-Encoder$^{M}$ | $8\times$ | – | **76.7** | 74.9 |
| *Llama3.1 8B decoder* | *closed-book* | $\infty$ | | 52.5 | 56.3 |
| | *open-book* | $1\times$ | | 84.0 | 77.2 |
| | ICAE-like$^{\dagger}$ | $\sim 32$ | 7.6B | 75.0 | 70.9 |
| | LLMLingua2 | $2.0\times$ | 0.6B | 73.4 | 75.1 |
| | | $4.0\times$ | – | 62.7 | 70.9 |
| | PISCO-like$^{\dagger}$ | $\sim 32$ | 7.6B | 62.9 | 67.8 |
| | | $4\times$ | – | 68.4 | 71.8 |
| | ARC$_4$-Encoder$^{\otimes}$ | $4\times$ | 3.0B | **81.1** | 76.3 |
| | ARC$_4$-Encoder$^{L}$ | $4\times$ | – | **81.1** | **77.1** |
| | ARC$_8$-Encoder$^{L}$ | $8\times$ | – | 80.2 | 75.5 |

## A.6 Context length performance scaling

In Fig. 7, we show how performance scales with the context length in the long-context setting. We address this by splitting QMSUM and QASPER into bins containing approximately the same number of samples. Each bin groups examples whose contexts fall within a specific token-length range. We see that independently of the task (question answering for QASPER and summarization for QMSUM), ARC-Encoder performance remains largely consistent across different token-length ranges. We attribute this context length robustness to our fine-tuning which contains samples of varying lengths.

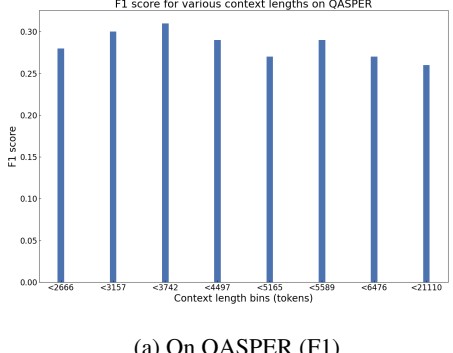
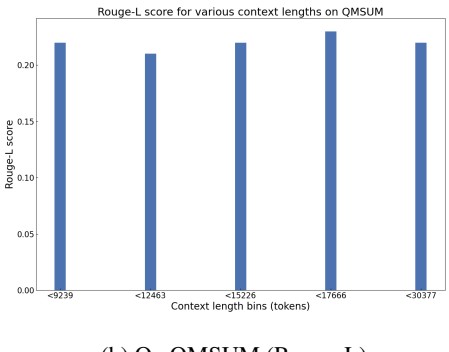

(a) On QASPER (F1)    (b) On QMSUM (Rouge-L)

Figure 7: **Performance scaling with context length**. On two benchmarks (a) QASPER and (b) QMSUM evaluation of our long-context ARC-Encoder paired with Llama2 7B Chat on subsets of the test dataset with similar context sizes.

## A.7 NEEDLE-IN-HAYSTACK ANALYSIS

Our method performs lossy context compression and our evaluation shows that this loss does not prevent the target LLM from extracting semantic information from the compressed representations for downstream tasks. To further explore this information loss, we evaluate fine-grained retrieval using the Needle-in-Haystack(gkamradt, 2023) (NIAH) benchmark in Fig. 8. In the following plots, we test ARC-Encoder (pooling factor 8) with Llama2 7B Chat in a long-context setting and compare it to other long-context-capable models, none of which were trained for this task. ARC-Encoder (bottom right) enables Llama2 7B Chat to retrieve precise information beyond its 4096-token window, and retrieval appears more consistent across varying context lengths and depths. Our model's top score is 7 because it outputs "Eat a sandwich and sit in Dolores Park on a sunny day" instead of the exact needle given to the LLM as a judge "The best thing to do in San Francisco is eat a sandwich and sit in Dolores Park on a sunny day." This demonstrates that ARC-Encoder enables fine-grained retrieval, but the model does not format the retrieved answer correctly.

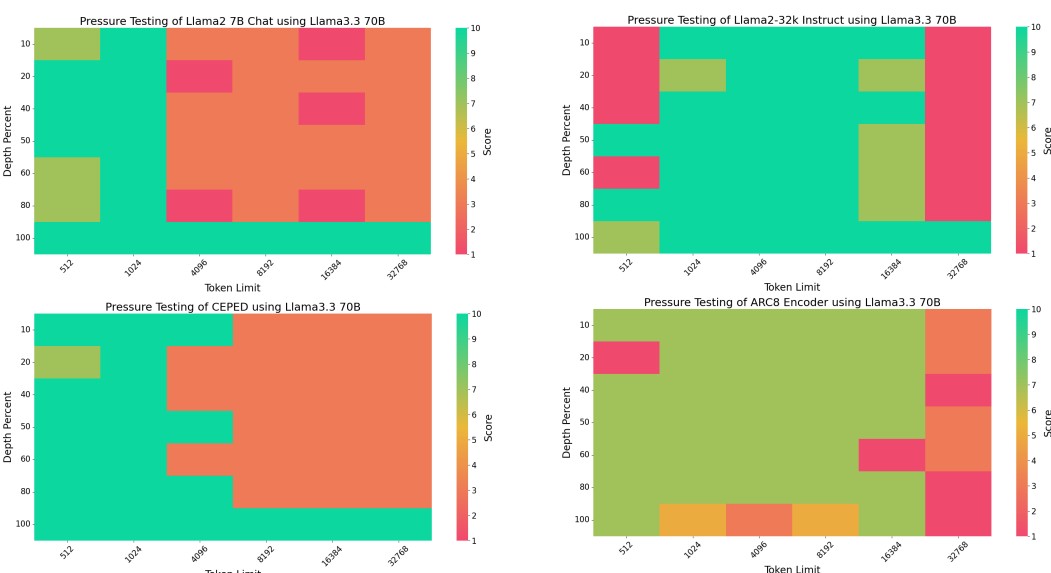

Figure 8: **Needle in the Haystack evaluation.** We test various models as in Tab. 3: Llama2 Chat (upper left), Llama2 32K fine-tuned for long-context (upper right), Llama2 with CEPED (bottom left) and our $ARC_8$-Encoder (bottom right).

## A.8 ON TARGETING A SMALLER LLM

An ARC-Encoder with 3B parameters does not offer efficiency gains when paired with smaller LLMs. However, our encoder ablations show that we can truncate more layers from the backbone or even switch to a different backbone to reduce its size. We therefore train two additional ARC-Encoder variants: one using Llama 3.2 3B truncated by 14 layers (1.8B parameters) and one using Llama 3.2 1B truncated by 2 layers (1.1B parameters). When combined with a 3B LLM (Llama 3.2 3B), both variants achieve strong performance as shown in Tab. 12, remaining close to the open-book setting, while also delivering computational benefits even at this scale, see Fig. 9 and Fig. 10. Our method should be adapted depending on the targeted LLM to preserve an effective performance–efficiency trade-off.

Table 12: **Smaller encoders paired with a smaller LLM.** Same setting and specifics as in Tab. 1 with Llama3.2 3B. We either use Llama3.2 3B with half of its layers truncated (1.8B parameters for the ARC-Encoder) or Llama3.1 1B truncated of 2 layers (1.1B parameters) as the backbone for an ARC-Encoder.

| Methods | PF | Param. | NQ | TQA | HQA | SQuAD | FLORES | CNN | Avg. |
|---|---|---|---|---|---|---|---|---|---|
| *closed-book* | $\infty$ | | 19.1 | 50.1 | 17.4 | 11.6 | | | |
| *open-book* | $1\times$ | | 34.4 | 65.5 | 43.2 | 71.4 | 29.3 | 26.0 | 45.0. |
| $ARC_4$-Encoder$^L$ | $4\times$ | 1.8B | 37.13 | 66.3 | 40.3 | 65.0 | 28.8 | 21.0 | 43.1 |
| $ARC_8$-Encoder$^L$ | $8\times$ | $-$ | 34.6 | 65.5 | 35.6 | 55.8 | 24.7 | 21.8 | 39.7 |
| $ARC_4$-Encoder$^L$ | $4\times$ | 1.1B | 34.9 | 65.9 | 37.9 | 63.4 | 27.6 | 20.8 | 41.8 |
| $ARC_8$-Encoder$^L$ | $8\times$ | $-$ | 33.0 | 64.4 | 33.2 | 52.4 | 23.0 | 20.5 | 37.8 |

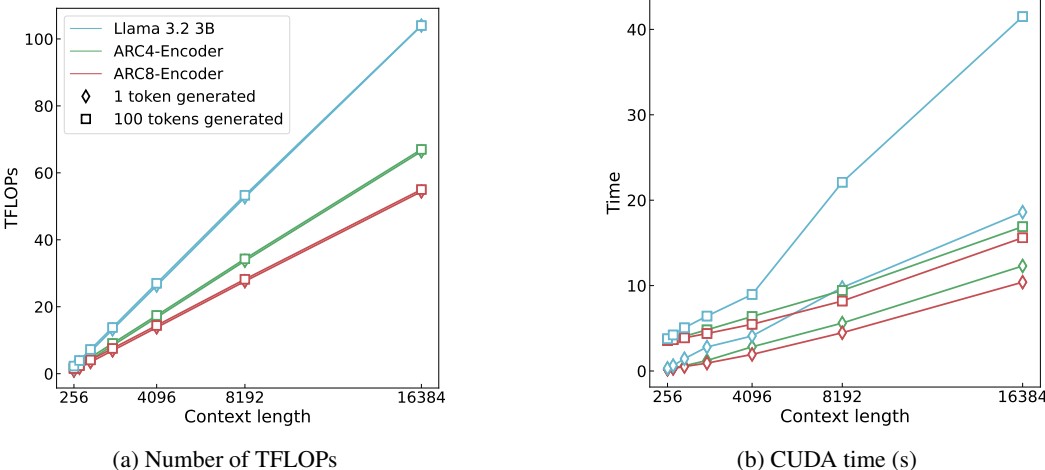

(a) Number of TFLOPs      (b) CUDA time (s)

Figure 9: **Measured computational costs for Llama 3.2 3B using ARC-Encoders with 1.8B parameters**. (a) Number of TFLOPs and (b) CUDA time in seconds for the continuation of a book from PG19 for various prompt lengths and numbers of tokens to generate on one NVIDIA H100.

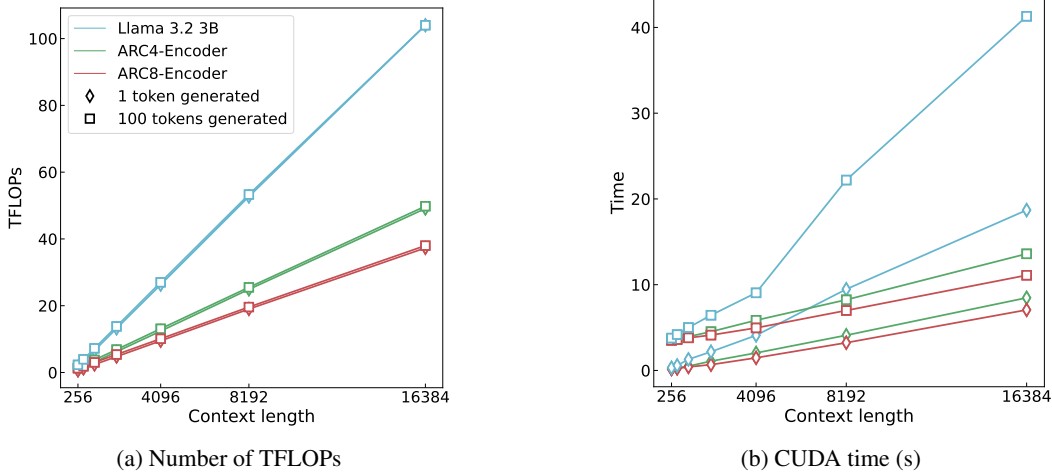

(a) Number of TFLOPs      (b) CUDA time (s)

Figure 10: **Measured computational costs for Llama 3.2 3B using ARC-Encoders with 1.1B parameters**. (a) Number of TFLOPs and (b) CUDA time in seconds for the continuation of a book from PG19 for various prompt lengths and numbers of tokens to generate on one NVIDIA H100.

# B    THEORETICAL GENERATION FLOPS

Let us denote $s$ the number of tokens of the prompt sequence, $d$ the hidden dimension of the model, $n_{\text{layers}}$ its number of layers, $v$ the vocabulary size and $N$ the overall number of parameters of the decoder. We assume for simplicity that the decoder uses multi-head attention and that the hidden dimension is the same everywhere in the model. The theoretical complexity of prefilling steps, measured in floating point operations (FLOPs) and neglecting norms, is:

- **Multi-head attention:**

$$\textbf{Q, K, V projections} = 3 \times s \times d^2$$
$$\textbf{Attention scores} = s^2 \times d$$
$$\times \textbf{V} = s^2 \times d$$
$$\textbf{Final projection} = s \times d^2 (\text{depends})$$
$$\textbf{Overall} = 4s \times d^2 + 2s^2 \times d$$

- **Feedforward network** $\propto s \times d^2$
- **Output projection** $\propto s \times d \times v$.

In the case where $d \gg s$, as the number of parameters per layer is proportional to $d^2$, we have $d \propto \sqrt{\frac{N}{n_{\text{layers}}}}$, which leads to:

$$\textbf{Total prefill FLOPs} \propto sN \quad \text{if } \sqrt{N} \gg s. \tag{1}$$

Hence, using $\frac{s}{x}$ tokens for prefilling instead of $s$ leads to a relative FLOPs of $\frac{1}{x}$.

At the opposite, when processing very large prompts, i.e, $d \ll s$, the computational complexity reads:

$$\textbf{Total prefill FLOPs} \propto s^2 \sqrt{N}. \tag{2}$$

If we use an ARC-Encoder, with a pooling factor of $x$, which has $n = p \times N$ parameters ($p < 1$) wih $p$ the relative size of the encoder vs. decoder, then the number of FLOPs to compress the prompt can be approximated to:

$$\begin{cases} \propto sN \times (p + \frac{1}{x}) & \text{if } d \gg s, \\ \propto s^2 \sqrt{N}(\sqrt{p} + \frac{1}{x^2}) & \text{if } d \ll s. \end{cases} \tag{3}$$

For instance, it is approximately $1.5\times$ smaller for ARC$_4$-Encoder and $1.9\times$ smaller for ARC$_8$-Encoder in the setting of the main table.

# C    TRAINING DETAILS

## C.1    DEFAULT SETTING

Trainings are performed on $8\times$H100 NVIDIA GPUs using PyTorch's FSDP framework[4]. The ablations follow the parameters and architectural choices from our best ARC$_8$-Encoder encoders 1, namely:

- **Encoder**: Llama3.2 3B truncated of the 2 last layers with every layer trained using a non-causal attention mask. When using Llama3.1 8B as encoder we truncate the 8 last layers.
- **Pooling**: by averaging queries in the last transformer block of the encoder with a pooling factor of 8.
- **MLP projector**: 2 learned matrices without activation function, with dimensions sequence $3072 \to 2048 \to 4096$ if the encoder is Llama3.2 3B, $4096 \to 2048 \to 4096$ otherwise.
- **Training**:
    - Special tokens are added depending on the task.

---

[4]https://docs.pytorch.org/docs/stable/fsdp.html

- – 20% reconstruction for pretraining during 60k steps with approximately 2B tokens seen by the encoder and maximum 256 tokens compressed and 256 tokens to continue or reconstruct.
- • During the continuation task, maximum 256 text tokens are prefixed to the compressed sequence to better align with the final few-shot evaluation.

- • **Fine-tuning**:
  - – `<Cont>` token is appended after each compressed sequence.
  - – Compressed representations are used as context and all the samples follow the format below (C.3) with more or less in-context examples as described in Tab. 15.
  - – Fine-tuning is performed with the same pooling factor as for the pretraining, unless stated otherwise.

**Remarks.** Inserting text tokens before the compressed sequence in the continuation task introduces a significant compute overhead during pretraining. Yet, after fine-tuning with interleaved few-shot samples, it offers no gains in the specific-decoder setting. Standard continuation can substitute for the "interleaved" one in pretraining, as long as fine-tuning later interleaves compressed and normal tokens. For consistency, we keep interleaved continuation in reported results since it helps ARC-Encoder generalize better in the multi-decoder setting.

## C.2 HYPERPARAMETERS

Table 13: Pretraining hyperparameters.

| Hyperparameters | Pretraining settings |
|---|---|
| optimizer | AdamW |
| max lr encoder | $1 \times 10^{-5}$ |
| max lr special tokens | $1 \times 10^{-5}$ |
| max lr rate MLP | $5 \times 10^{-5}$ |
| lr scheduler type | 1 cycle policy |
| init lr | $1 \times 10^{-20}$ |
| final lr | $1 \times 10^{-10}$ |
| warmup steps | 1000 |
| weight decay | 0.1 |
| batch size | 16 |
| gradient accumulation steps | None |
| GPUs | 8 |
| max tokens compressed at once | 256 |
| prefix text tokens | $\leq 256$ |
| max norm (gradient clipping) | 1.0 |
| mixed precision | Yes |
| number of steps | 60k for ablations |
| | 80k for multi-decoder |
| | 80k for ARC-Encoder (long context or not) |
| MLP init. | Kaiming Unif. leaky-ReLU slope of $\sqrt{5}$ |
| Special tokens init. | Ones |

**Multi-Decoder specificities.** To add OLMo 7B to the existing multi-decoder ARC$_4$-Encoder, we fine-tune the MLP using 8k steps with 100 warmup steps and a maximum learning rate for the MLP of $10^{-4}$.

## C.3 CONTEXT COMPRESSION FINE-TUNING

To generate synthetic fine-tuning data, we use the vLLM[5] library for fast, efficient single-GPU inference. Gemma 3 27B (Team et al., 2025) is then prompted to generate translations of Atlas

---

[5] https://docs.vllm.ai

Table 14: Fine-tuning hyperparameters. Not listed hyperparameters are identical to pretraining ones.

| Hyperparameters | Context Compression | Long Context |
|---|---|---|
| max lr encoder | $2 \times 10^{-6}$ | − |
| max lr special tokens | $2 \times 10^{-6}$ ($3 \times 10^{-5}$ for multi-decoder) | NA |
| max lr rate MLP | $3 \times 10^{-5}$ | − |
| final lr | $1 \times 10^{-8}$ | − |
| warmup steps | 50 | 1000 |
| weight decay | $5 \times 10^{-2}$ | − |
| n. steps | 4k (8k for multi-decoder) | 8k |
| batch size | 8 | 2 |
| gradient accumulation steps | None | 4 |
| max tokens compressed at once | 2048 | 1024 |
| interleaved examples | $\leq$ # in-context examples ($M$) | NA |
| | $= M$ for Tabs. 1, 2, 6 and 9 | NA |
| contexts chunked to | No | 1024 tokens |
| max contexts in parallel | 1 | 31 |

Wikipedia passages (up to 3 concatenated passages) in various languages that we split into two categories:

1) Spanish, French, German and Danish;

2) Hindi, Russian, Swahili, Arabic, Turkish, Japanese, Finnish and Chinese (simplified).

We mix these generated translation datasets with supervised QA, summarization and reading comprehension datasets. For QA datasets, we retrieve the top-5 passages using NV-Embed (Lee et al., 2025), based on Wikipedia sequences[6] from KILT framework (Petroni et al., 2021). To improve information retrieval in large compressed contexts, we concatenate retrieved passages when possible and compress them jointly. The number of concatenated passages is randomly sampled between 1 and the maximum number of retrieved passages. This is particularly beneficial for HotpotQA and CNN, with minimal impact on other evaluation benchmarks. For MS MARCO (Nguyen et al., 2016) we removed the samples without answers (*no answer*). The complete list of the subsets that make up our final fine-tuning dataset is reported in Tab. 15, along with the proportions according to which these subsets are sampled from.

| Subsets | Proportion | # in-context examples | Max. concat. passages |
|---|---|---|---|
| Synth. translations | | | |
| Group 1 translations | 6% | 5 | |
| Group 2 translations | 4% | 5 | |
| QA with retrieved context | | | |
| AdversarialQA (Bartolo et al., 2020) | 8% | 5 | 4 |
| FreebaseQA (Jiang et al., 2019) | 27% | 5 | 4 |
| ASQA (Stelmakh et al., 2023) | 1% | 5 | 4 |
| MS MARCO (Nguyen et al., 2016) | 9% | 5 | 4 |
| SciQ (Johannes Welbl, 2017) | 3% | 5 | 4 |
| Reading comprehension | | | |
| DROP (Dua et al., 2019) | 20% | 5 | |
| ParaSCI (Dong et al., 2021) | 12% | 5 | |
| Summarization | | | |
| DialogSum (Chen et al., 2021) | 2.5% | 3 | |
| SAMSum (Gliwa et al., 2019) | 2.5% | 4 | |
| WikiSum (Cohen et al., 2021) | 5% | 5 | |

Table 15: Fine-tuning dataset for context compression

---

[6]https://huggingface.co/datasets/dmrau/kilt-128

---

**Few-shot compressed fine-tuning template**

Document: `<TOKENS_COMPRESSED>`
Question: `<QUESTION>`
Answer: `<ANSWER>`

Document: `<TOKENS_COMPRESSED>`
Question: `<QUESTION>`
Answer: `<ANSWER>`
⋮
Document: `<TOKENS_COMPRESSED>`
Question: `<QUESTION>`
Answer: `<ANSWER>`                    (*the loss is computed only on the last answer*)

---

## C.4 LONG CONTEXT

### C.4.1 PRETRAINING

To pretrain ARC-Encoder when paired with an instruct decoder model, we continue to alternate between continuation and reconstruction tasks. We format each of the pretraining samples using the following template C.4.1.

---

**Pretraining templates with Llama2 Chat decoder**

**Template:**
`` [INST] Prefix+`<TOKENS_COMPRESSED>`+Instruction [/INST] Suffix ``
Reconstruction:

- Prefix = `"Text:\n\n"`
- Instruction = `"\n Replicate the input text."`
- Suffix = `"Replicated text:\n"`

Continuation:

- Prefix = `"Text:\n\n"`
- Instruction = `"\n Continue the previous text."`
- Suffix = `"Text continuation:\n..."`

---

### C.4.2 FINE-TUNING

For this part, we synthesized QA, summarization and paraphrasing examples using the same procedure as in Appendix C.3.

**QA generation.** We split books from PG-19 (Rae et al., 2019) and arXiv papers from RedPajama (Weber et al., 2024) into paragraphs, randomly selecting 5 consecutive paragraphs. We then prompted Gemma3-27B to generate questions and gold answers using instructions such as:

- *As a human instructor assessing students' comprehension of a scientific article, you craft a concise question that ideally requires a short phrase or sentence to answer. If the article lacks the necessary information, the answer should be 'unanswerable'. For yes/no questions, reply with 'yes', 'no', or 'unanswerable'. Then, supply the gold answer.*
- *You are given a story from a book. Your task is to create a question that can be answered in a short phrase or sentence. Then, provide the gold answer.*

The final context in the dataset is the entire book or paper. Additionally, we generate QA examples from Wikipedia by selecting one chunk from the Atlas Wikipedia dataset and appending up to 20 chunks from the same source before prompting the model.

**Summarization generation.** Using the same datasets, we split the texts into 10 groups of passages. Gemma3 27B is prompted to summarize each subsection; then, based on these summaries, it is prompted again to produce a higher-level summary, with prompt variations controlling the target length. For Atlas Wikipedia, we directly ask the model to produce a short summary from 10 consecutive chunks. In both QA and summarization tasks, we truncate contexts longer than 500k characters and discard those shorter than 1k characters.

**Paraphrase.** To mimic the questions asked in QM-Sum benchmarks, we prompt Gemma3 to reformulate passages of the text. For all tasks, we truncate contexts longer than 500k characters and discard those shorter than 1k characters.

| Contexts | # samples | Mean Ctx | Median Ctx | Mean answer |
|---|---|---|---|---|
| Summarization | | | | |
| From Atlas | 64000 | 6833 | 5593 | 714 |
| From PG-19 books | 64000 | 55048 | 34915 | 635 |
| From ArXiv papers | 40000 | 10896 | 4875 | 834 |
| Paraphrase | | | | |
| From PG-19 books | 80000 | 31854 | 29272 | 665 |
| From ArXiv papers | 64000 | 5625 | 4260 | 617 |
| QA | | | | |
| From Atlas | 80000 | 8065 | 8325 | 43 |
| From PG-19 books | 80000 | 317179 | 322942 | 69 |
| From ArXiv papers | 40000 | 54526 | 43386 | 49 |

Table 16: **Fine-tuning dataset statistics for long-context understanding**. We report different statistics on the length in characters of the contexts ('Ctx') and the answers for each subset of fine-tuning samples.

All samples are inserted in an instruction prompt depending on their task and context dataset using the same template as in C.4.1 with adapted *Prefix*, *Instruction* and *Suffix*.

# D  EVALUATION DETAILS

## D.1  BASELINES IMPLEMENTATION

**LLMLingua2 (Pan et al., 2024):** We used the open-source model *microsoft/llmlingua-2-xlm-roberta-large-meetingbank* following the instructions from LLMLingua.

**xRAG (Cheng et al., 2024):** We use the official codebase from xRAG and extend it to support Llama3.1 8B as a decoder. Due to architectural similarities between Mistral 7B and Llama3.1 8B, only minor modifications are required. We first pretrain the MLP projector by closely following the instructions and data from the repository and the original paper, adapting it to base models by removing all chat templates. Next, we modify the fine-tuning dataset pre-processing to interleave compressed context in an ICL-style format, aligned with our fine-tuning template (see C.3), which closely matches the evaluation setup. To ensure consistency and avoid variability due to dataset size or quality, we use our own dataset for fine-tuning. We observe in Tabs. 1 and 9 that xRAG performs poorly on translations tasks (FLORES). After further investigations, we believe that compressing the full sequence into one vector leads to a loss of information that causes partial-only translations or hallucinations, as illustrated below.

**ICAE (Ge et al., 2024):** For this re-implementation, we use our own codebase. We follow the hyperparameters and design choices described in the original paper, including the use of special tokens and alternating pretraining tasks. We set the language modeling task ratio to 0.5 and pretrain the encoder on our evaluated decoders (Llama3.1 8B and Mistral 7B) with our crawl dataset for 100k steps (which is half the number of steps reported in the paper, but the training curve had already converged). Additionally, we adapt the fine-tuning template to match our evaluation format (see Appendix C.3). To avoid redundancy with our ablation studies on pooling methods with memory

tokens, we retain the fine-tuning dataset from Ge et al. (2024) (PwC), which was specifically synthesized for this purpose. However, due to the poor generalization on our evaluation benchmarks, we present ICAE-like models fine-tuned on our own dataset.

**PISCO (Louis et al., 2025a):** As with ICAE, we re-implemented PISCO using our own codebase, following the hyperparameters and design choices outlined in the original paper (e.g., LoRA applied to both encoder and decoder, encoder architecture, use of memory tokens). Since the official code is not publicly available, we referred to Ge et al. (2024) for implementation details not specified in the paper, such as the use of special tokens. To ensure consistency and avoid variability from dataset quality, we use our custom fine-tuning dataset but increased the number of fine-tuning steps to 8000 to reach near 500k samples which matches the number of training samples used in the PISCO paper. Furthermore, our custom dataset consists in PISCO fine-tuning dataset without the train sets of the evaluation datasets, with extra summarization datasets and synthesized translations data. While sequence-level distillation is key to avoiding reliance on gold labels, early experiments with silver labels showed that using gold labels enables a fairer comparison. Additionally, we train variants that process fixed-length input chunks of size 128 tokens, enabling a fixed pooling factor and aligning the setup more closely with that of Louis et al. (2025a).

---

**xRAG with Mistral 7B failures to translate English texts**

**To French:**

- `Ground-truth:` "Cette page est accessible facilement à partir d'une seule adresse Web, ce qui la rend facile à mémoriser et à écrire pour les étudiants qui ne savent pas utiliser un clavier et qui ont des problèmes d'orthographe."

- `xRAG generation:` "Avec un seul nom de domaine, il est facile pour les utilisateurs d'accéder à l'information, ce qui est un avantage pour les étudiants."

**To Spanish:**

- `Ground-truth:` "Son superiores a los servidores proxy por varios motivos: redirigen todo el tráfico de Internet y no únicamente los http."

- `xRAG generation:` "Estos son más eficientes que los proxy, ya que no requieren que el usuario realice cambios en sus configuraciones de red."

**To German:**

- `Ground-truth:` "Vergessen Sie nicht die Extrakosten für weitere Visa, Abfluggebühren, Transportmittel an Land etc. für all die Orte außerhalb von Afrika mit einzuberechnen."

- `xRAG generation:` "Das ist zwar teurer als die Flüge, aber das Geld ist es wert, weil man damit nicht nur das Flugzeug spart, sondern auch die Kosten für die Übernachtung, die Verpflegung und die Reiseversicherung."

---

## D.2 EVALUATION DATASETS

**Context Compression.** We evaluate our pipeline on question answering (QA) and reading comprehension tasks using the following benchmarks: HotpotQA (Yang et al., 2018) (*distractor* setting on the dev set, 7400 samples) , Natural Questions (Kwiatkowski et al., 2019) (NQ open dev set, 3605 samples), TriviaQA (Joshi et al., 2017) (unfiltered nocontext validation set, 11308 samples), SQuAD (Rajpurkar et al., 2016) (10565 samples). When ground-truth context is not provided, we retrieve the top-5 passages using NV-Embed (Lee et al., 2025), based on Wikipedia sequences from the Atlas framework (Izacard et al., 2022), effectively simulating a RAG setup. The number of retrieved passages used for evaluation is specified for each benchmark. We report Exact Match (EM) as our primary evaluation metric, where answers are normalized and EM = 1 if all characters match exactly. We demonstrate summarization capabilities on the CNN-DailyMail dataset (a subset of 1000 samples of the dev set), evaluating performance with the Rouge-L metric, as Zhang et al. (2023) noted that strong Rouge-L scores in this context are closely aligned with high human approval.

For translation tasks, we evaluate on the FLORES benchmark (Goyal et al., 2021) (992 samples), using BLEU scores computed with SacreBLEU[7]. BLEU scores are averaged over four translation directions: English to Danish, French, German, and Spanish. Models are prompted in a 5-shot setting using the following template, using compressed contexts for each example. Examples are sampled from the validation set and are fixed among all models. The reported pooling factor reflects the average per-context compression of tokens, not the ratio over the full prompt (including the textual prompt). It consists in dividing the number of tokens of the full document using the decoder tokenizer by the number of compressed tokens or the number of tokens of the compressed document in the hard compression case.

---

**Evaluation QA template**

- **n examples for $n$-shot evaluation:**
  Document: `<TOKENS_COMPRESSED>`
  Question: `<QUESTION>`
  Answer: `<ANSWER>`

  Document: `<TOKENS_COMPRESSED>`
  Question: `<QUESTION>`
  Answer: `<ANSWER>`
  ⋮

- **the final question**
  Document: `<TOKENS_COMPRESSED>`
  Question: `<QUESTION>`
  Answer:

---

**Evaluation translation template**

- **n examples for $n$-shot evaluation:**
  Document: `<TOKENS_COMPRESSED>`
  Question: Translate the previous document into `<LANGUAGE>`.
  Answer: `<ANSWER>`
  ⋮

- **the final question**
  Document: `<TOKENS_COMPRESSED>`
  Question: Translate the previous document into `<LANGUAGE>`.
  Answer:

---

**Long Context.** For long-context understanding, we report results on NarrativeQA (NQA), QASPER (Qspr), GovReport (GvRp), and QM-Sum validation datasets from ZeroSCROLLS (Shaham et al., 2023) benchmark, a suite of zero-shot long-context understanding tasks that emphasize instruction-following capabilities. We evaluate on the full validation dataset which consists in respectively 3461, 1726, 973 and 272 samples. Specifically, we adopt the task formats and instructions as used in Yen et al. (2024)[8].

---

[7]https://github.com/mjpost/sacrebleu
[8]https://github.com/princeton-nlp/CEPE

