# OpenReview forum: "ARC-Encoder: learning compressed text representations for large language models"
_ICLR.cc/2026/Conference — Submitted to ICLR 2026_

### Official Review · Reviewer_6Bmd · 2025-10-19

**Soundness:** 3
**Presentation:** 3
**Contribution:** 3
**Rating:** 4
**Confidence:** 4

**Summary:**

The authors proposed a new context compressor, ARC-Encoder to reduce the size of context representations. ARC-Encoder can generalize to many LLMs and show advanced performance on downstream tasks.

**Strengths:**

1. The paper is well-written and easy to follow. The motivation of this paper is strong and practical.
2. The method formulation is clear and reasonable.
3. Authors conduct a comprehensive evaluation on many downstream tasks to demonstrate the effectiveness and generalizability of the proposed method.

**Weaknesses:**

1. Despite of the effectiveness, the idea of context compression has been provided in many previous works, some of which are not included in the referece, e.g., [1][2][3]. This impairs the novelty of the prompsed method.
2. A big issue that resides in context compression is inevitable information loss, especially when being tested on fine-grained retrieval task, such as Needle-in-Haystack (NIAH). The authors should provide more results on these types of tasks to demonstrate their model's  capability in handling fine-grained retrieval tasks.



[1] Chevalier, Alexis, et al. "Adapting language models to compress contexts." arXiv preprint arXiv:2305.14788 (2023).

[2] Han, Wei, et al. "Two are better than one: Context window extension with multi-grained self-injection." arXiv preprint arXiv:2410.19318 (2024).

[3] Zhang, Peitian, et al. "Soaring from 4k to 400k: Extending llm’s context with activation beacon." arXiv preprint arXiv:2401.03462 2.3 (2024): 5.

**Questions:**

See weakness

---

> ### Author Response · Authors · 2025-11-18
>
> First, we want to thank the reviewer for their thorough evaluation of our work and their insightful remarks.
>
>
> &nbsp;
>
>
> ### W1. On the novelty of our method.
> &nbsp;
>  We add these references to the related work section of the updated submission (Context Compression and Long Context). First, [1] relies on additional learned tokens (summary vectors) appended to the sequence, similar to ICAE, which always produce a fixed number of outputs regardless of input length. In contrast, our pooling-based approach allows a fixed pooling factor, and our comparisons with ICAE-like models show that ARC-Encoder achieves stronger performance. Secondly, we propose a similar encoder-decoder architecture as in [2]. However, we insert the compressed representations in the text stream of the decoder as [2] uses cross-attention to condition the generation. This enables us to perform in-context learning using compressed representations while keeping the decoder unmodified and thus undamaged. Moreover, we do not rely on a similar splitting and context tree strategy, our encoder learns directly how to compress information so the decoder can use it across tasks, simplifying the overall pipeline. Finally, Activation Beacon [3] extends context length by inserting activation “anchors” into the model’s hidden states, but it requires modifying and fine-tuning the target LLM itself; in contrast, ARC-Encoder leaves the decoder untouched and provides portable compressed representations that can be plugged into different LLMs without architectural changes.
>
>
> &nbsp;
>
>
> ### W2. About tests on fine-grained retrieval tasks.
> &nbsp;
>  As the reviewer mentions, our method performs lossy context compression. In Appendix A.7, we test ARC-Encoder (pooling factor 8) with Llama2 7B Chat in a long-context setting and compare it to other long-context-capable models, none of which were trained for this task. ARC-Encoder enables Llama2 7B Chat to retrieve precise information beyond its 4096-token window, and retrieval appears more consistent across varying context lengths and depths than with any other tested baseline.  Our model’s top score is 7 because it outputs "Eat a sandwich and sit in Dolores Park on a sunny day" instead of the exact needle given to the LLM as a judge "The best thing to do in San Francisco is eat a sandwich and sit in Dolores Park on a sunny day." This demonstrates that ARC-Encoder enables fine-grained retrieval, but the model does not format the retrieved answer correctly. We add these new experiments in the Appendix A.7 (page 20) of the revised submission.  Please note that the main motivation for our method are tasks requiring information integration/aggregation over a long context, as opposed to tasks focused on recalling a precise and local fact, such as NIAH, where retrieval-based methods might be better suited.
>
>
> &nbsp;
>
>
> ### References
>
> [1] Chevalier, Alexis, et al. "Adapting language models to compress contexts." arXiv preprint arXiv:2305.14788 (2023).
>
>
> [2] Han, Wei, et al. "Two are better than one: Context window extension with multi-grained self-injection." arXiv preprint arXiv:2410.19318 (2024).
>
>
> [3] Zhang, Peitian, et al. "Soaring from 4k to 400k: Extending llm’s context with activation beacon." arXiv preprint arXiv:2401.03462 2.3 (2024): 5.
>
>
> [4] Tao Ge, Jing Hu, Lei Wang, Xun Wang, Si-Qing Chen, & Furu Wei. (2024). In-context Autoencoder for Context Compression in a Large Language Model.

---

> > ### Comment · Reviewer_6Bmd · 2025-11-26
> > **Response to authors**
> >
> > Thanks for the response. In my view, using fixed [1] or variable number of tokens does not differ a lot. I also do not agree with the opinion of "damage" since in [2][3] they retrain the decoder to adapt to the new learning pattern. Authors should justify more why training the encoder only can do better or comparable to joint training approach.
> >
> > For W2, the fine-grained retrieval capability is greatly important as most of the compression-based approaches have address this issue.

---

> > > ### Author Response · Authors · 2025-12-02
> > >
> > > We thank the reviewer for the follow-up.
> > >
> > > &nbsp;
> > >
> > > ## On the distinction between pooling and fixed number of compressed tokens.
> > >
> > > &nbsp;
> > >
> > > We agree that the number of compressed tokens alone does not determine novelty. Our point is that different compression mechanisms offer distinct trade-offs. First, as shown in our pooling ablations (Section 4.4, line 451), per-segment pooling consistently outperforms memory tokens (as in [1]) in our experiments. Second, ARC-Encoder’s pooling provides a tunable compression factor that naturally scales with input length and enables evaluation or fine-tuning at multiple compression ratios. In contrast, extending memory-token methods to lower compression rates typically requires additional chunking heuristics, which reduces flexibility.
> > >
> > > &nbsp;
> > >
> > > ##  On encoder-decoder joint training.
> > >
> > > &nbsp;
> > >
> > > We acknowledge that our earlier use of the term “damage” may be imprecise. On the one hand, [2] and [3] focus on context extension experimenting especially in long-context settings. On the other hand, ARC-Encoder necessitates **fewer hyperparameters** to tune (one compression ratio instead of one per node level) and provides efficiency benefits along with strong performance in the **short- and long-context regime**. [3] does not demonstrate efficiency improvements below the model’s native context window focusing solely on long-context scenarios. Therefore, ARC-Encoder benefits from broader applications and use-cases. Importantly, unlike our approach, [2] and [3] modify and retrain parts of the decoder. In [2], as far as we understand all tokens must pass through newly trained tree cross-attention, which may introduce distribution shift and catastrophic forgetting [5] on out-of-distribution inputs.
> > > Freezing the decoder provides new opportunities of applications:
> > > - **Encoder sharing across decoders:** a single ARC-Encoder can be reused with multiple LLMs (Section 4.2), whereas joint training would require separate fine-tuned encoder-decoder pairs leading to much more specific parameters.
> > > - **Reusable compressed representations:** fixed decoders allow storing and reusing compressed representations with low memory cost (Section 4.5) for various downstream tasks. This is difficult for [2], where compressed chunks depend on the query, and for [3], where dimension compression is not straightforward.
> > >
> > >
> > > &nbsp;
> > >
> > > In summary, while joint encoder–decoder training as in [2] and [3] can outperform encoder-only training on certain specialized downstream tasks, our approach offers a more flexible and generalizable alternative: a plug-and-play,  compression module that preserves the behavior of pretrained LLMs. It avoids catastrophic forgetting risks associated with decoder modifications, and unlocks new use cases.
> > >
> > > &nbsp;
> > >
> > > ## About W2.
> > >
> > > We thank the reviewer for this helpful remark. Fine-grained retrieval is an important aspect of compression-based approaches. To improve the presentation of our method, we have added a brief discussion in Section 4.3 (line 395) highlighting that ARC-Encoder achieves more consistent retrieval performance across varying context lengths and depths than other tested baselines.
> > >
> > > &nbsp;
> > >
> > > ### References
> > > [1] Chevalier, Alexis, et al. "Adapting language models to compress contexts." arXiv preprint arXiv:2305.14788 (2023).
> > > [2] Han, Wei, et al. "Two are better than one: Context window extension with multi-grained self-injection." arXiv preprint arXiv:2410.19318 (2024).
> > > [3] Zhang, Peitian, et al. "Soaring from 4k to 400k: Extending llm’s context with activation beacon." arXiv preprint arXiv:2401.03462 2.3 (2024): 5.
> > > [4] Tao Ge, Jing Hu, Lei Wang, Xun Wang, Si-Qing Chen, & Furu Wei. (2024). In-context Autoencoder for Context Compression in a Large Language Model.
> > > [5] Yun Luo, Zhen Yang, Fandong Meng, Yafu Li, Jie Zhou, & Yue Zhang. (2025). An Empirical Study of Catastrophic Forgetting in Large Language Models During Continual Fine-tuning.

---

### Official Review · Reviewer_5MmT · 2025-10-26

**Soundness:** 3
**Presentation:** 3
**Contribution:** 2
**Rating:** 4
**Confidence:** 4

**Summary:**

This paper proposes ARC-Encoder, a method that compresses input text into compact continuous representations to reduce context length for LLMs. It achieves efficient inference without changing the model (decoder) architecture.

**Strengths:**

1. The proposed ARC-Encoder offers a solution for context compression, achieving this without altering the underlying LLM architecture.

2. The method demonstrates strong empirical results over different tasks.

**Weaknesses:**

1. The claim that ARC-Encoder “works seamlessly with multiple LLMs” is overstated, since in practice it still requires fine-tuning separate projectors for each target model, even if the number of parameters is small.

2. The encoder model is very large (e.g., ~3B parameters), which raises serious concerns about practical efficiency. The paper should provide detailed FLOPs and latency analyses to substantiate efficiency claims.

3. It is unclear how the text encoder’s embeddings are initialized, given that different LLMs have distinct embedding spaces. Without careful alignment, transferring across models could lead to catastrophic forgetting or representation collapse.

4. Training such a large encoder (e.g., 3B) to replace token embeddings in decoder LLMs is resource-intensive, involving multiple training stages (pretraining, finetuning, multi-decoder adaptation). The paper should clarify why this approach is preferable to soft compression methods that require LLM fine-tuning.

5. It would further strengthen the paper if the authors could expand the related work discussion to include a recent study that adopts vision encoder for token compression [ref1]. Incorporating this perspective would help situate ARC-Encoder more comprehensively within the evolving landscape of encoder-based compression approaches and clarify its unique contributions.

[ref1] Vision-centric Token Compression in Large Language Model (NeurIPS 2025)

**Questions:**

1. Are the fine-tuning datasets and downstream evaluation datasets strictly out-of-domain, ensuring a fair assessment of generalization?

2. In the long-context experiments, what is the maximum context length tested, and how does performance scale with length?

3. Does ARC-Encoder also provide benefits in short-context scenarios, or is its advantage limited to long-context settings?

---

> ### Author Response · Authors · 2025-11-18
> **Answer to the weaknesses**
>
> First, we want to thank the reviewer for their thorough evaluation of our work and their insightful remarks.
>
>
> &nbsp;
>
>
> ### W1. On our overstatement in the abstract.
> &nbsp;
> We acknowledge that our statement may be misleading and change the last sentence of the abstract, in the revised submission, by: “This makes ARC-Encoder a flexible and efficient solution for portable encoders that can support multiple LLMs, requiring only small model-specific projectors for adaptation.”
>
>
> &nbsp;
>
>
> ### W2. About  details of FLOPs and latency gains.
> &nbsp;
> These kinds of experiments were formerly in the Appendix (B.2) of the submission. We move it to the main part of the revised submission (Section 4.6, page 9) to highlight the computational and latency benefits of compressing the prefilling context with our method. As shown in Figure 5, generation is less costly in terms of compute when using compressed tokens by ARC-Encoder. The compute cost of compression is amortized during the prefilling phase since the decoder has fewer tokens to process.
>
>
> &nbsp;
>
>
> ### W3. About the initialization of the encoder embeddings.
> &nbsp;
>  We understand that we should better explain how we design our text encoder. ARC-Encoders are trained starting from any decoder-only LLM (typically Llama 3.2 3B) with removed layers and bidirectional attention. As the reviewer notes, the resulting hidden states do not initially lie in the target decoder’s embedding space. The MLP head addresses this by projecting pooled encoder representations into the decoder’s embedding space, and this alignment is learned during pretraining via the reconstruction task. In practice, early training updates mainly reduce the reconstruction loss, suggesting that the encoder and MLP first learn to produce correctly aligned compressed representations, after which the continuation objective encourages representations that the decoder can use effectively for downstream generation.  We add a few lines in the preprint (lines 417-420) to shortly clarify this point.
>
>
> &nbsp;
>
>
> ### W4. On the advantages of our method compared to other soft compression methods that require LLM fine-tuning:
>   - With the proposed approach, LLMs’ performance remains unchanged, preserving the exact behaviors and evaluations reported by its creators. In contrast, fine-tuning the LLM on compressed representations often degrades performance on standard tokens or necessitates the use of multiple sets of weights at inference, complicating pipelines where compressed and full-text inputs are interleaved.
>   - With a single set of decoder weights, our approach allows feeding both raw text and ARC_x-Encoder compressed representations (for any pooling factor x the encoder was trained for).
>   - In principle, ARC-Encoders could even be trained on black-box LLMs using only outputs and gradients, leaving the decoder weights untouched.
>
> &nbsp;
> Furthermore, as the reviewer notes, training ARC-Encoders involves multiple stages. However, most soft-compression methods that fine-tune the LLM also require an initial pretraining phase (e.g., OSCAR [1], GMSA [2]); to our knowledge, only PISCO [3] avoids it. The mentioned approaches likewise train encoders of similar scale to ours. We add a few lines in the revised submission (lines 96-97) to clarify this point.
>
>
> &nbsp;
>
>
> ### W5. Reference to compression through vision encoder.
> &nbsp;
> In the revised submission, we expand the related-work discussion to integrate [5]. The paper adopts an encoder–decoder architecture with cross-attention instead of interleaving pooled embeddings with full text. While this reduces computation, we found it severely harms few-shot performance in our setting. To obtain meaningful pooling, the authors also rely on Probability-Informed Visual Enhancement to preserve semantic consistency through text anchoring. In contrast, because our encoder operates directly in the text space, no additional objectives are required, resulting in a simpler and more streamlined method.

---

> > ### Author Response · Authors · 2025-11-18
> > **Answer to the questions**
> >
> > ### Q1. On ensuring a fair assessment of generalization.
> > &nbsp;
> >  When designing our fine-tuning dataset, we ensured that no training examples were from the benchmarks we evaluated on (both training and test sets), though most tasks still remain in domain. Motivated by the reviewer’s remark, we conducted additional experiments to assess strict out-of-domain generalization by evaluating our models on BioASQ [6] and PubMedQA [7]. These two QA datasets focus on biological and biomedical question-answering. In Table 11 in the Appendix A.5.2 of the submission,  these new results highlight strong transferability: despite all compressor baselines (except LLMLingua2) being trained on the same data, our ARC-Encoder achieves the best performance, even at a pooling factor of 8, reaching scores close to the upper-bound open-book setting and demonstrating robust real-world applicability. We add these new experiments in the Appendix A.5.2 (page 18)  of the revised submission.
> >
> >
> > &nbsp;
> >
> >
> > ## Out-of-domain evaluations (F1 scores)
> > _Same setting and specifics as in Table 1 of the submission._
> >
> > ### **Mistral 7B Decoder**
> >
> > | Method | PF | Params | PubMedQA | BioASQ |
> > |-------|----|--------|----------|--------|
> > | *closed-book* | ∞ | – | *58.7* | *63.6* |
> > | *open-book* | 1× | – | *84.4* | *77.5* |
> > | ICAE-like† | ~32 | 7.2B | 63.6 | 57.7 |
> > | LLMLingua2 | 1.9× | 0.6B | 74.4 | 73.4 |
> > |  | 3.6× | – | 66.4 | 69.7 |
> > | PISCO-like† | ~32 | 7.2B | 62.4 | 60.5 |
> > |  | 4× | – | 62.0 | 66.0 |
> > | **ARC_4-Encoder^⊗** | 4× | 3.0B | 73.9 | 72.0 |
> > | **ARC_4-Encoder^M** | 4× | – | 76.6| **75.6** |
> > | **ARC_8-Encoder^M** | 8× | – | **76.7** | 74.9 |
> >
> > ---
> >
> > ### **Llama 3.1 8B Decoder**
> >
> > | Method | PF | Params | PubMedQA | BioASQ |
> > |--------|----|--------|----------|--------|
> > | *closed-book* | ∞ | – | *52.5* | *56.3* |
> > | *open-book* | 1× | – | *84.0* | *77.2* |
> > | ICAE-like† | ~32 | 7.6B | 75.0 | 70.9 |
> > | LLMLingua2 | 2.0× | 0.6B | 73.4 | 75.1 |
> > |  | 4.0× | – | 62.7 | 70.9 |
> > | PISCO-like† | ~32 | 7.6B | 62.9 | 67.8 |
> > |  | 4× | – | 68.4 | 71.8 |
> > | **ARC_4-Encoder^⊗**  | 4× | 3.0B | **81.1** | 76.3 |
> > | **ARC_4-Encoder^L** | 4× | – | **81.1** | **77.1** |
> > | **ARC_8-Encoder^L** | 8× | – | 80.2| 75.5 |
> >
> >
> >
> > &nbsp;
> >
> >
> > ### Q2. On the performance scaling with context length.
> > &nbsp;
> > In the long-context experiments, to align with the other tested models, we restrict ourselves to context lengths of 32 768 tokens. However, ARC-Encoders can accept larger contexts since they are based on Llama 3.2 3B which has a context window limit of 128k. We add an experiment to show how performance scales with length by splitting QMSUM and QASPER into bins containing approximately the same number of samples. This experiment is in the revised submission in Appendix A.6, page 19. Each bin groups examples whose contexts fall within a specific token-length range. We see that independently of the task (question answering for QASPER and summarization for QMSUM), ARC-Encoder performance remains largely consistent across different token-length ranges. We attribute this context length robustness to our fine-tuning which contains samples of varying lengths.
> >
> >
> >
> > &nbsp;
> >
> >
> > ### Q3. About short-context scenarios benefits.
> > &nbsp;
> >  ARC-Encoders achieve strong downstream performance on shorter-context tasks, such as Natural Questions and TriviaQA (~180 tokens, Table 1 in the submission). Additionally, Figure 5 page 10  in our revised submission shows measurable FLOPs and latency improvements even for contexts as short as 256 tokens.
> >
> >
> > &nbsp;
> >
> >
> > ### References
> >
> > [1] Maxime Louis, Thibault Formal, Hervé Dejean, & Stéphane Clinchant. (2025). OSCAR: Online Soft Compression And Reranking.
> >
> > [2] Jiwei Tang, Zhicheng Zhang, Shunlong Wu, Jingheng Ye, Lichen Bai, Zitai Wang, Tingwei Lu, Jiaqi Chen, Lin Hai, Hai-Tao Zheng, & Hong-Gee Kim. (2025). GMSA: Enhancing Context Compression via Group Merging and Layer Semantic Alignment.
> >
> > [3] Maxime Louis, Hervé Déjean, & Stéphane Clinchant. (2025). PISCO: Pretty Simple Compression for Retrieval-Augmented Generation.
> >
> > [4] Jesse Mu, Xiang Lisa Li, & Noah Goodman. (2024). Learning to Compress Prompts with Gist Tokens.
> > [5] Ling Xing, Alex Jinpeng Wang, Rui Yan, Xiangbo Shu, & Jinhui Tang. (2025). Vision-centric Token Compression in Large Language Model.
> >
> > [6] Krithara, A., Nentidis, A., Bougiatiotis, K., & Paliouras, G. (2023). BioASQ-QA: A manually curated corpus for Biomedical Question Answering. Scientific Data, 10.
> >
> > [7] Qiao Jin, Bhuwan Dhingra, Zhengping Liu, William W. Cohen, & Xinghua Lu. (2019). PubMedQA: A Dataset for Biomedical Research Question Answering.

---

> > > ### Comment · Reviewer_5MmT · 2025-11-23
> > >
> > > 1. Figure 5 only reports results based on Mistral-7B. We therefore continue to raise serious concerns about the practical efficiency of the approach across different LLM scales. Since the encoder itself is very large (3B), it substantially increases the overall model size. It remains unclear whether ARC-ENCODER would still provide any efficiency benefits when applied to smaller LLMs such as 1B–3B models.
> > >
> > > 2. Regarding the response to W5, is the statement “*because our encoder operates directly in the text space, no additional objectives are required*” accurate? The hidden states of ARC-ENCODER do not initially lie in the target decoder’s embedding space, and the paper addresses this through an additional reconstruction task.

---

> ### Author Response · Authors · 2025-12-02
>
> We thank the reviewer for the follow-up.
>
> &nbsp;
>
> ## Q1. On the practical efficiency of ARC-Encoders when applied to smaller LLMs.
>
> &nbsp;
>
> An ARC-Encoder with 3B parameters would indeed not yield efficiency gains when paired with LLMs smaller than 3B. However, our encoder ablations show that we can substantially reduce encoder size by truncating more layers from the backbone or by switching to a smaller backbone entirely.
> To demonstrate this, we train two additional ARC-Encoder variants:
> - a 1.8B encoder obtained by removing 14 layers from Llama 3.2 3B,
> - a 1.1B encoder obtained by removing 2 layers from Llama 3.2 1B.
>
> When paired with a 3B LLM (Llama 3.2 3B), both variants maintain strong performance, remaining close to the open-book setting, while also providing computational gains, as shown in Figures 9 and 10. These results illustrate that the method can be adapted to different LLM sizes to preserve an effective performance–efficiency trade-off. We include these new experiments in Appendix A.8 (page 21) of the revised submission.
>
> &nbsp;
>
> ### Smaller encoders paired with a small LLM
>
> &nbsp;
>
> Same setting as Table 1 with Llama 3.2 3B.
>
>
>
> | Methods             | PF | Param. | **NQ** | **TQA** | **HQA** | **SQuAD** | **FLORES** | **CNN** | **Avg.** |
> | ------------------- | -- | ------ | ------ | ------- | ------- | --------- | ---------- | ------- | -------- |
> | *closed-book*       | ∞  | –      | 19.1   | 50.1    | 17.4    | 11.6      | –          | –       | –        |
> | *open-book*         | 1× | –      | 34.4   | 65.5    | 43.2    | 71.4      | 29.3       | 26.0    | 45.0     |
> | **ARC_4-Encoder^L** | 4× | 1.8B   | 37.1   | 66.3    | 40.3    | 65.0      | 28.8       | 21.0    | 43.1     |
> |  **ARC_8-Encoder^L**        | 8× | –      | 34.6   | 65.5    | 35.6    | 55.8      | 24.7       | 21.8    | 39.7     |
> |  **ARC_4-Encoder^L**   | 4× | 1.1B   | 34.9   | 65.9    | 37.9    | 63.4      | 27.6       | 20.8    | 41.8     |
> |  **ARC_8-Encoder^L**      | 8× | –      | 33.0   | 64.4    | 33.2    | 52.4      | 23.0       | 20.5    | 37.8     |
>
>
> &nbsp;
>
> ## Q2. Regarding our response to W5.
>
> &nbsp;
>
> The reviewer is correct that the encoder’s hidden states do not initially lie in the decoder’s embedding space, and we agree that our phrasing was imprecise. Our intention was to highlight that ARC-Encoder does not require explicit alignment to text-token embeddings. Instead, it uses a reconstruction objective that teaches the decoder to interpret these new compressed representations and a continuation objective to promote interpretable representations for downstream tasks . This approach allows the model to determine, during training, which information should be preserved in the compressed representation. In contrast, methods such as [5] rely on a preprocessing step, selecting lower-frequency tokens, to define which information is retained.

---

### Official Review · Reviewer_kLrY · 2025-10-30

**Soundness:** 3
**Presentation:** 3
**Contribution:** 2
**Rating:** 4
**Confidence:** 4

**Summary:**

This paper proposes ARC-Encoder, which adopts an architecture consisting of an "encoder (based on Llama3.2 3B, with the output head and causal mask removed) + a 2-layer MLP projector". It performs average pooling on consecutive queries in the last self-attention layer (with a compression factor of 4/8) to generate continuous representations that replace token embeddings in the decoder, without modifying the decoder. The training process employs alternating pretraining tasks of "reconstruction-continuation" and task-specific fine-tuning, enabling a single encoder to adapt to multiple decoders (with dedicated MLP parameters accounting for < 1%). Experiments verify that ARC-Encoder achieves performance close to the open-book baseline on QA, translation, summarization, and long-context tasks. Additionally, the storage size of compressed Wikipedia representations is comparable to that of raw text. Its core contributions lie in high compatibility, multi-decoder adaptation, and long-context extension.

**Strengths:**

ARC-Encoder does not require decoder modification, enabling adaptation to existing LLMs. For multi-decoder adaptation, only a small amount of parameters are needed, resulting in low deployment costs. It covers both short- and long-context tasks, and memory analysis supports precomputation, indicating great potential for practical application.

**Weaknesses:**

It has weak innovation: its framework is highly similar to ICAE, and there are no breakthrough designs in multi-decoder adaptation or long-context strategies. Furthermore, it fails to explore performance at high compression factors (16×/32×) and generalization in professional domains, nor does it provide comparisons of inference latency in real-world scenarios.

**Questions:**

Have the authors compared the performance of ARC-Encoder with similar context compression architectures like ICAE under the same settings? What are the core technical differences between them?

---

> ### Author Response · Authors · 2025-11-18
>
> First, we want to thank the reviewer for their thorough evaluation of our work and their insightful remarks.
>
>
> &nbsp;
>
>
> ### Q1. About comparisons of ARC-Encoders with ICAE like architectures and higher compression rates:
> &nbsp;
> ICAE-style compression uses learned memory tokens appended to the sequence, producing a fixed number of compressed vectors per input. In contrast, our method requires no extra tokens: the encoder directly pools the input, generating a variable number of compressed representations that scale with sequence length and yield consistent performance across context sizes. Moreover, ICAE encoders replicate the architecture of their target decoder, whereas ARC-Encoders are derived from any decoder-only LLM (e.g., Llama 3.2 3B) by removing layers and adding bidirectional attention, making them significantly smaller and more flexible.
> In Table 1, we compare our models to an “ICAE-like” baseline, which adapts ICAE to our datasets and training setup. To enable comparisons at fixed pooling rates, we also train and evaluate ICAE architectures on fixed-size chunks following Section 3.3.3 of the ICAE paper [1], giving them a fixed pooling factor. Table 10 in Appendix A.5.1 (page 18) reports these results. **Under the same settings, our method consistently outperforms ICAE-like models at matched pooling factors while requiring less than twice the encoding compute, and on some benchmarks even surpasses them when using ARC-Encoders with larger pooling factors.**
> We would also like to show that ARC-Encoders can be explored at higher pooling factors. As expected, performance degrades as the pooling factor increases. However, **even at a pooling factor of 32, our model outperforms significantly the closed-book baseline** proving that the decoder can extract meaningful semantic information from these highly compressed representations. We add these new experiments as well as clarifications of the core technical differences in the Appendix A.5.1 (page 18) of the revised version of our submission.
>
> &nbsp;
>
> ### Further baselines evaluations
> _Same setting as Table 1 (Mistral 7B decoder)._
> Pooling factors for each benchmark which output a fixed number of tokens appear in parenthesis.
>
> | Method | PF | NQ | TQA | HQA | SQuAD | FLORES | CNN | Avg. |
> |-------|----|----|-----|-----|-------|--------|-----|------|
> | *closed-book* | ∞ | **29.1** | **62.4** | **22.8** | **17.1** | – | – | – |
> | *open-book* | 1× | **39.9** | **70.5** | **48.3** | **77.7** | **31.3** | **27.2** | **49.2** |
> | ICAE-like† | ~32 | 36.5 (5×) | 66.7 (5×) | 24.3 (46×) | 58.8 (6×) | 28.3 (1×) | 15.8 (32×) | 38.4 |
> |  | 4× | 36.4 | 66.7 | 23.8 | 60.5 | 28.7 | 18.6 | 39.1 |
> |  | ~16 | 35.7 (10×) | 66.7 (9×) | 26.0 (92×) | 51.0 (12×) | 26.9 (2×) | 14.3 (64×) | 20.6 |
> |  | 8× | 34.8 | 66.6 | 8.9 | 53.0 | 26.7 | 17.5 | 34.6 |
> |  | ~8 | 34.9 (19×) | 65.3 (19×) | 25.3 (185×) | 28.9 (23×) | 18.5 (4×) | 17.7 (128×) | 18.1 |
> |  | 16× | 33.7 | 65.6 | 0.1 | 31.1 | 19.0 | 14.7 | 27.4 |
> | **ARC_2-Encoder^M** | 2× | **41.7** | **69.3** | **48.3** | **76.7** | 30.4 | 18.9 | **47.6** |
> | **ARC_4-Encoder^M** | 4× | 39.0| 68.9 | 45.1 | 71.1 | **31.0** | **23.8** | 46.5 |
> | **ARC_8-Encoder^M** | 8× | 38.4 | 67.9 | 40.8 | 62.0 | 28.3 | 22.9 | 43.4 |
> | **ARC_16-Encoder^M** | 16× | 35.4 | 67.1 | 31.8 | 45.1 | 22.7 | 20.3 | 37.1 |
> | **ARC_32-Encoder^M** | 32× | 34.6 | 65.8 | 28.8 | 34.8 | 17.0 | 17.8 | 33.1 |

---

> > ### Author Response · Authors · 2025-11-18
> > **Rest of the first answer**
> >
> > ### W1. About generalization of our method in professional domains and performance at higher compression factors.
> > &nbsp;
> > In the following table (Appendix Table 11), we evaluate models on PubMedQA [2] and BioASQ [3], two biomedical QA datasets absent from our fine-tuning corpus, enabling a strict out-of-domain test on professional domains. All compressor baselines, except LLMLingua2, were fine-tuned on the same dataset. Despite this, our ARC-Encoder attains the best performance, even with a pooling factor of 8, approaching the upper-bound open-book setting. These results highlight the robustness and real-world applicability of our method, and we include the experiments in Appendix A.5.2 (page 18) of the revised submission.
> >
> >
> > &nbsp;
> >
> >
> > ### Professional and out-of-domain evaluations (F1 scores)
> > _Same setting and specifics as in Table 1 of the submission._
> >
> > ### **Mistral 7B Decoder**
> >
> > | Method | PF | Params | PubMedQA | BioASQ |
> > |-------|----|--------|----------|--------|
> > | *closed-book* | ∞ | – | *58.7* | *63.6* |
> > | *open-book* | 1× | – | *84.4* | *77.5* |
> > | ICAE-like† | ~32 | 7.2B | 63.6 | 57.7 |
> > | LLMLingua2 | 1.9× | 0.6B | 74.4 | 73.4 |
> > |  | 3.6× | – | 66.4 | 69.7 |
> > | PISCO-like† | ~32 | 7.2B | 62.4 | 60.5 |
> > |  | 4× | – | 62.0 | 66.0 |
> > | **ARC_4-Encoder^⊗** | 4× | 3.0B | 73.9 | 72.0 |
> > | **ARC_4-Encoder^M** | 4× | – | 76.6| **75.6** |
> > | **ARC_8-Encoder^M** | 8× | – | **76.7** | 74.9 |
> >
> > ---
> >
> > ### **Llama 3.1 8B Decoder**
> >
> > | Method | PF | Params | PubMedQA | BioASQ |
> > |--------|----|--------|----------|--------|
> > | *closed-book* | ∞ | – | *52.5* | *56.3* |
> > | *open-book* | 1× | – | *84.0* | *77.2* |
> > | ICAE-like† | ~32 | 7.6B | 75.0 | 70.9 |
> > | LLMLingua2 | 2.0× | 0.6B | 73.4 | 75.1 |
> > |  | 4.0× | – | 62.7 | 70.9 |
> > | PISCO-like† | ~32 | 7.6B | 62.9 | 67.8 |
> > |  | 4× | – | 68.4 | 71.8 |
> > | **ARC_4-Encoder^⊗**  | 4× | 3.0B | **81.1** | 76.3 |
> > | **ARC_4-Encoder^L** | 4× | – | **81.1** | **77.1** |
> > | **ARC_8-Encoder^L** | 8× | – | 80.2| 75.5 |
> >
> >
> > &nbsp;
> >
> >
> > ### W2. Inference latency in real-world scenarios.
> > &nbsp;
> > We also believe that comparisons of inference latency and computation profiling in real-world scenarios would benefit our presentation. These kinds of experiments were formerly in the Appendix (B.2) of the submission. We move them to the main part of the new version of the submission (Section 4.6, page 9)  to show the computational and latency benefits of compressing the prefilling context with our method. As shown in the figure, generation is less costly in terms of compute when using compressed tokens by ARC-Encoder. The compute cost of compression is amortized during the prefilling phase since the decoder has fewer tokens to process.
> >
> >
> > &nbsp;
> >
> >
> > ### References
> >
> > [1] Tao Ge, Jing Hu, Lei Wang, Xun Wang, Si-Qing Chen, & Furu Wei. (2024). In-context Autoencoder for Context Compression in a Large Language Model.
> >
> >
> > [2] Qiao Jin, Bhuwan Dhingra, Zhengping Liu, William W. Cohen, & Xinghua Lu. (2019). PubMedQA: A Dataset for Biomedical Research Question Answering.
> >
> > [3] Krithara, A., Nentidis, A., Bougiatiotis, K., & Paliouras, G. (2023). BioASQ-QA: A manually curated corpus for Biomedical Question Answering. Scientific Data, 10.

---

### Official Review · Reviewer_darH · 2025-10-31

**Soundness:** 3
**Presentation:** 2
**Contribution:** 3
**Rating:** 8
**Confidence:** 3

**Summary:**

This paper proposes ARC-Encoder, a plug-and-play encoder that compresses textual context into continuous representations for frozen LLM decoders. The approach aims to reduce inference cost while maintaining downstream performance, without modifying the decoder architecture. The work is comprehensive and empirically strong, with experiments spanning multiple decoders, tasks, and compression factors.

**Strengths:**

1. This paper introduces a new formulation of context compression that does not alter the decoder. Unlike prior “memory token” or “gist token” methods, ARC-Encoder performs fixed-ratio pooling within the encoder’s attention layers and connects to decoders through a lightweight MLP. This architectural separation is elegant and conceptually clean.
2. The authors conduct a broad and fair evaluation across multiple domains. Results show consistent improvements over strong baselines, often matching or surpassing open-book settings despite heavy compression.
3. The analytical experiments are extensive, and ablations on pretraining tasks, pooling strategies, and encoder truncation are well thought out.

**Weaknesses:**

1. This paper does not provide a deeper theoretical discussion of why pooled query averaging in attention preserves semantic fidelity or why it outperforms token-level compression. A brief analytical or representational argument could strengthen the paper’s foundation.
2. How sensitive is performance to the dimensionality of the MLP bottleneck?

**Questions:**

See above.

---

> ### Author Response · Authors · 2025-11-18
>
> First, we want to thank the reviewer for their thorough evaluation of our work and their insightful remarks.
>
>
> &nbsp;
>
>
> ### W1. About a more theoretical discussion of the pooling.
> &nbsp;
> We observed  that applying token-level compression after the final self-attention (SA) gives similar results. We prefer pooled query averaging because it is slightly cheaper computationally. However, it also outperforms token-level compression before the last SA layer.  The reason is representational: pooling tokens before the last SA forces pooled queries to attend to pooled keys, producing linear combinations of pooled values, effectively mixing information too early. In contrast, query averaging uses pooled queries that still attend to full token-level keys and values, allowing each pooled representation to be formed as a direct linear combination of token-level representations. This keeps pooled vectors in the same embedding space as the backbone, simplifying training and enabling the use of a small MLP for projection into the decoder’s embedding space.
> In Appendix A.3, we further show that an ARC-Encoder built on Llama 3.1 8B (to match hidden dimensions) can be trained to feed exactly the same compressed representations to both Llama 8B and Mistral 7B. Although downstream performance is lower than with decoder-specific MLPs, it still surpasses the closed-book baseline, demonstrating that both decoders can extract semantic information from these pooled representations, which lie in an embedding space between those of Mistral 7B and Llama 8B.
>
> &nbsp;
>
> ### W2. On the sensitivity of our models to the dimensionality of the MLP bottleneck.
> &nbsp;
> In Section 4.5 (memory analysis), we assess how performance varies with the dimensionality of the MLP bottleneck. Although the experiment also includes quantization, the conclusions regarding bottleneck size hold even without quantization. In Figure 4, the bottleneck dimension varies within curves of the same color and the marker shape indicates the pooling factor. With PF = 4, the MLP bottleneck can be reduced much further without harming performance. We clarify the ablation in Section 4.5 (lines 476-478) of the revised preprint.

---

### Author Response · Authors · 2025-11-18

We thank all reviewers for their thorough and constructive evaluations. A revised version of the submission incorporating their suggestions is now uploaded (modifications are in blue), including:
- Moving the section about  the **computational and latency benefits of our method** through profiling from the appendix to the main text (Section 4.6).
- Adding new experiments showing strong results of our models on out-of-domain benchmarks highlighting **generalization to out-of-domain** and specialized professional domains, in Appendix A.5.2.
- Adding experiments about sensitivity to the context length in Appendix A.6 and performing **evaluation on fine-grained retrieval tasks** (Needle-in-Haystack)  in Appendix A.7.
- Clarifying the advantages of our approach compared to other soft compression methods in the related work and in Appendix A.5.1.
- Adding new references in the related work section to better situate ARC-Encoder and clarify its innovative aspect.
- Writing minor additional explanations of our ablations and design choices in the core paper.


&nbsp;


We appreciate that reviewers found our experimental results interesting, and we acknowledge their request for greater clarity on the novelty of our approach. To address this, we performed additional experiments and expanded our explanations to better demonstrate the method’s benefits.

---

> ### Author Response · Authors · 2025-12-02
> **Summary of the improvements (1/3)**
>
> We thank the reviewers for their evaluations and for the constructive discussions that helped improve the paper. Given the particular format of this rebuttal, and to make the previous exchanges easier to follow, we group and summarize below the main weaknesses and questions raised by the reviewers, and highlight how we address each of these concerns.
>
> &nbsp;
>
> ___
> ### 1. On the generalizability of ARC-Encoders.
>
> &nbsp;
>
> **Reviewers concerns.**
> Reviewers asked for stronger evidence of generalization in more challenging settings: evaluation on out-of-domain data (6Bmd and kLrY), performance at higher compression ratios, analysis of the MLP bottleneck size (darH), and consistency across context lengths in long-context scenarios (5MmT).
>
> **Our response.**
> While the paper already included experiments showing generalizability across datasets, by testing on benchmarks not included in our fine-tuning dataset, the reviewers’ comments motivated additional experiments in more challenging settings. First, we evaluate strict out-of-domain generalization by testing our models on BioASQ [6] and PubMedQA [7]. These two datasets focus on biological and biomedical question-answering which are out-of-domain in comparison to our fine-tuning dataset.  As shown in Table 11 (Appendix A.5.2), ARC-Encoder consistently outperforms other compressor baselines, even at a pooling factor of 8, reaching scores close to the open-book upper bound. Second, we also explored higher compression levels and found that even at pooling factors of 32, ARC-Encoder still substantially outperforms the closed-book baseline (for instance +100% on SQuAD), demonstrating that the decoder can recover meaningful semantics from heavily compressed representations (Appendix A.5.1). Finally, we evaluated robustness to context length by grouping QMSUM and QASPER examples by token range (Appendix A.6) and observed stable performance across bins, which we attribute to fine-tuning on samples of diverse lengths. Finally, in Section 4.5 we clarify how performance varies with the size of the MLP bottleneck, and in Appendix A.7 we show that ARC-Encoder maintains consistent fine-grained retrieval accuracy across context lengths under lossy compression through a Needle-in-a-Haystack analysis.

---

> > ### Author Response · Authors · 2025-12-02
> > **Summary of the improvements (2/3)**
> >
> > ### 2. On the unique contribution of our method.
> >
> > &nbsp;
> >
> > **Reviewers concerns.**
> > Reviewers requested further clarification on how ARC-Encoder differs from existing context compression methods. In particular, reviewer kLrY asked for a stronger comparison with approaches using memory tokens, such as ICAE [1]. Reviewer 5MmT noted that recent method [2] employs vision encoders to compress textual context, and that discussing them would help highlight our unique contribution. Similarly, reviewer 6Bmd suggested including references [3,4,5] to better substantiate the novelty of our approach.
> >
> >
> > **Our response.**
> > With ARC-Encoder, our objective is to develop a flexible, plug-and-play context compressing mechanism. This focus leads to several contributions that distinguish our work from prior methods.
> > First, we keep the decoder LLM entirely unchanged, unlike approaches such as [2, 4, 5], which train additional cross-attention layers in the decoder. Modifying the decoder in this way can introduce distribution shifts and increase the risk of catastrophic forgetting on out-of-distribution inputs. By freezing the decoder, ARC-Encoder allows encoder-only fine-tuning without using any examples from the training set of the evaluation benchmarks, while still achieving competitive or superior results compared to recent methods [7, 8].
> > Freezing the decoder also enables two practical use cases:
> > - Encoder sharing across decoders, allowing the same ARC-Encoder to work with multiple LLMs (Section 4.2), unlike approaches requiring joint training.
> > - Reusable compressed representations that can be stored once and efficiently reused for different downstream tasks (Section 4.5).
> >
> > Second, ICAE-style compression uses learned memory tokens appended to the sequence, producing a fixed number of compressed vectors per input. In contrast, ARC-Encoder introduces no additional tokens: the encoder directly pools the input, generating a variable number of compressed representations that scale with sequence length and yield consistent performance across context sizes. Under the same settings, our method consistently outperforms ICAE-like models at matched pooling factors while requiring less than twice the encoding compute, and on some benchmarks even surpasses them when using ARC-Encoders with larger pooling factors (Table 1 and Table 10). As shown in our pooling ablations (Section 4.4, line 451), in our specific setting, with equivalent encoder architecture, per-segment pooling consistently outperforms memory tokens.
> > Finally, we reduce the number of hyperparameters and training stages to enhance reproducibility. Unlike [2, 4], we do not rely on additional preprocessing steps. Instead, the model learns in a unified pre-training setup where continuation and reconstruction are alternated, followed by encoder-only fine-tuning.
> >
> > Together, these design choices constitute the core novelty of ARC-Encoder. They enable unique applications while maintaining strong performance and delivering computational benefits. ARC-Encoder is built for flexibility and plug-and-play deployment. We plan to release pre-trained ARC-Encoders so that users can accelerate inference by feeding the decoder with ARC-Encoder representations, without requiring any modifications to their LLM.

---

> ### Author Response · Authors · 2025-12-02
> **Summary of the improvements (3/3)**
>
> ### 3. On the analysis of efficiency gains.
>
> &nbsp;
>
> **Reviewers concerns.**
> Reviewers kLrY and  5MmT asked for detailed FLOPs and latency analysis.
>
> **Our response.**
> We previously reported these results in Appendix B.2, but we move them to the main submission (Section 4.6, page 9) to better highlight the computational and latency benefits of compressing the prefilling context. As shown in Figure 5, decoder generation becomes cheaper when using ARC-Encoder compressed tokens, and the compression cost is amortized since the decoder processes fewer tokens. To demonstrate efficiency across model sizes, we introduce two smaller ARC-Encoder variants, one based on a truncated Llama 3.2 3B (1.8B parameters) and one on a truncated Llama 3.2 1B (1.1B parameters), which are both smaller versions of ARC-Encoders. Combined with a 3B LLM, both variants maintain performance close to the open-book setting while still offering computational gains. This shows that ARC-Encoder can be adapted to different LLM scales to preserve an effective performance–efficiency trade-off. These new experiments are included in Appendix A.8 (page 21).
>
> &nbsp;
>
> ___
> To conclude, the reviewers’ feedback strengthened both our results and their presentation. In the revised submission, we clarify several experiments and results, moving them from the appendix into the main paper, expand the related work to better situate our contributions, and add new experiments that further demonstrate the strengths of ARC-Encoders in more challenging scenarios.
>
>
> &nbsp;
>
> ### References:
> [1] Tao Ge, Jing Hu, Lei Wang, Xun Wang, Si-Qing Chen, & Furu Wei. (2024). In-context Autoencoder for Context Compression in a Large Language Model.
> [2] Ling Xing, Alex Jinpeng Wang, Rui Yan, Xiangbo Shu, & Jinhui Tang. (2025). Vision-centric Token Compression in Large Language Model.
> [3] Chevalier, Alexis, et al. "Adapting language models to compress contexts." arXiv preprint arXiv:2305.14788 (2023).
> [4] Han, Wei, et al. "Two are better than one: Context window extension with multi-grained self-injection." arXiv preprint arXiv:2410.19318 (2024).
> [5] Zhang, Peitian, et al. "Soaring from 4k to 400k: Extending llm’s context with activation beacon." arXiv preprint arXiv:2401.03462 2.3 (2024): 5.
> [6] Krithara, A., Nentidis, A., Bougiatiotis, K., & Paliouras, G. (2023). BioASQ-QA: A manually curated corpus for Biomedical Question Answering. Scientific Data, 10.
> [7] Qiao Jin, Bhuwan Dhingra, Zhengping Liu, William W. Cohen, & Xinghua Lu. (2019). PubMedQA: A Dataset for Biomedical Research Question Answering.
> [8] Jiwei Tang, Zhicheng Zhang, Shunlong Wu, Jingheng Ye, Lichen Bai, Zitai Wang, Tingwei Lu, Jiaqi Chen, Lin Hai, Hai-Tao Zheng, & Hong-Gee Kim. (2025). GMSA: Enhancing Context Compression via Group Merging and Layer Semantic Alignment.
> [9] Maxime Louis, Hervé Déjean, & Stéphane Clinchant. (2025). PISCO: Pretty Simple Compression for Retrieval-Augmented Generation.

---

### Meta-Review · Area_Chair_i87E · 2025-12-21

**Summary:**

This paper proposes ARC-Encoder, a plug-and-play encoder that compresses long textual contexts into continuous representations consumable by frozen LLM decoders. The approach is clearly motivated and empirically extensive, with experiments across multiple decoders, tasks, compression ratios, and context lengths. Reviewers generally agree that the method is sound, well written, and practically motivated.

However, the decision is primarily shaped by concerns about the level of novelty and the strength of the conceptual contribution, rather than by correctness or experimental rigor. While the rebuttal significantly strengthens the empirical story and clarifies design choices, it does not fully resolve doubts about whether ARC-Encoder represents a sufficiently distinct advance over prior context compression and context-extension approaches.

**Reviewer Concerns:**

**Concerns largely addressed by the rebuttal**

(1) The authors add extensive new experiments, including out-of-domain biomedical QA, higher compression factors (up to 32×), robustness across context lengths, fine-grained retrieval (Needle-in-a-Haystack), and efficiency profiling. These additions substantially improve the paper’s empirical coverage.

(2) Moving FLOPs/latency analysis into the main text and adding smaller encoder variants paired with smaller LLMs addresses several practicality concerns.

(3) Direct ICAE-like baselines under matched settings and expanded related-work discussion clarify empirical differences and improve contextualization.

(4) The authors appropriately soften language (e.g., “seamless” multi-LLM support) and clarify alignment, training, and ablation details.



**Concerns that remain outstanding**

(1) Despite clarifications, multiple reviewers remain unconvinced that the core idea—encoder-based context compression with pooled representations and frozen decoders—constitutes a sufficiently new conceptual contribution relative to prior work on memory tokens, context autoencoders, and encoder–decoder or joint-training approaches. The differences are seen as largely architectural and design-choice driven, rather than introducing a new principle or paradigm.

(2) While the rebuttal articulates practical advantages (portability, reuse, avoiding distribution shift), at least one reviewer remains unconvinced that encoder-only training is fundamentally preferable or competitive in principle compared to joint encoder–decoder training, especially for tasks requiring precise information retention.

(3) Although new Needle-in-a-Haystack results are provided, reviewers note that fine-grained retrieval remains a known weakness of compression-based methods and that ARC-Encoder does not clearly overcome this limitation beyond being competitive with other lossy approaches.

Overall, the remaining concerns are conceptual rather than empirical: the paper demonstrates that the method works, but some reviewers are not persuaded that it advances the state of the art in a sufficiently novel way.

**Reviewer Scores:**

Reviewer darH: Likely unchanged at 8 (accept); concerns were minor and well addressed.

Reviewer kLrY: Unlikely to change score, acknowledging stronger comparisons and added experiments, but still viewing the contribution as incremental.

Reviewer 5MmT: Likely to move from 4 to 6,  remaining skeptical about overall cost–benefit and positioning.

Reviewer 6Bmd: Likely to remain around 4, as concerns about novelty and the choice of encoder-only training versus joint training are only partially resolved.

---

### Decision · Program_Chairs · 2026-01-26

Reject